# EGFR-dependent TOR-independent endocycles support *Drosophila* gut epithelial regeneration

Jinyi Xiang[1,2], Jennifer Bandura[1,2], Peng Zhang[1,2,3], Yinhua Jin[1,2], Hanna Reuter[4] & Bruce A. Edgar[1,2,3]

Following gut epithelial damage, epidermal growth factor receptor/mitogen-activated protein kinase (EGFR/MAPK) signalling triggers *Drosophila* intestinal stem cells to produce enteroblasts (EBs) and enterocytes (ECs) that regenerate the gut. As EBs differentiate into ECs, they become postmitotic, but undergo extensive growth and DNA endoreplication. Here we report that EGFR/RAS/MAPK signalling is required and sufficient to drive damage-induced EB/EC growth. Endoreplication occurs exclusively in EBs and newborn ECs that inherit EGFR and active MAPK from fast-dividing progenitors. Mature ECs lack EGF receptors and are refractory to growth signalling. Genetic tests indicated that stress-dependent EGFR/MAPK promotes gut regeneration via a novel mechanism that operates independently of Insulin/Pi3K/TOR signalling, which is nevertheless required in nonstressed conditions. The *E2f1* transcription factor is required for and sufficient to drive EC endoreplication, and Ras/Raf signalling upregulates *E2f1* levels posttranscriptionally. We illustrate how distinct signalling mechanisms direct stress-dependent versus homeostatic regeneration, and highlight the importance of postmitotic cell growth in gut epithelial repair.

[1] German Cancer Research Center (DKFZ), Im Neuenheimer Feld 280, 69120 Heidelberg, Germany. [2] Center for Molecular Biology of The University of Heidelberg (ZMBH), Im Neuenheimer Feld 282, 69120 Heidelberg, Germany. [3] Huntsman Cancer Institute, University of Utah, 2000 Circle of Hope, Salt Lake City, Utah 84108, USA. [4] Department of Stem Cells and Regeneration, Max Planck Institute for Molecular Biomedicine, Röntgenstraße 20, 48149 Münster, Germany. Correspondence and requests for materials should be addressed to B.A.E. (email: bruce.edgar@hci.utah.edu).

Cells are adept at altering their function to adapt to environmental changes. One major type of adaptation, hyperplasia (increased cell number), is commonly observed in diverse species and tissues, and has been studied extensively in various physiological and pathological contexts. Another important adaption, cellular hypertrophy (increased cell size), is observed for instance in muscle following increased nutrition or exercise. However, the control of cell size *in vivo* in response to stress is not well studied.

The endodermal portion of the *Drosophila* intestine, the midgut, is a good model for hypertrophic cell adaptation. The midgut is maintained by intestinal stem cells (ISCs). ISC divisions generate enteroblasts (EBs), the nondividing progenitors that differentiate into diverse cell types falling into two of the major classes, namely enterocytes (ECs) and enteroendocrine cells[1,2]. ECs are large, absorptive polyploid cells that constitute >90% of the mass of the midgut. Enteroendocrine cells are small diploid secretory cells[3,4]. ISCs and EBs express *escargot* (*esg*), a transcription factor specific to progenitor cells. Furthermore, the intestinal epithelium is surrounded by a sheath of visceral muscle that comprises an important component of the ISC niche[5]. Under normal physiological conditions the midgut epithelium is continuously renewed by stem cell activity, with complete epithelial replacement taking 14–21 days. However, following epithelial damage or stress, such as from bacterial infection, gut epithelial self-renewal can accelerate dramatically to maintain homeostasis. Recent studies demonstrate that many signalling pathways regulate midgut renewal to maintain homeostasis. For example Delta/Notch signalling is required for EB to EC differentiation. ISCs specifically express the Notch ligand, *Delta* (*Dl*)[6]. A Notch activity reporter gene, *Su(H)GBE*, containing binding sites for the Notch-responsive transcription factor Su(H) is expressed specifically in *Dl*-negative *esg*-positive EBs, indicating that Notch signalling is active in these cells[7]. JAK/STAT (Janus kinase/signal transducer and activator of transcription) signalling also plays an important role in the tissue's damage response: damaged ECs secrete cytokines called Unpaireds (*Upd2*, *Upd3*) that activate JAK/STAT signalling in ISC and EB cells, promoting their proliferation and differentiation to replace lost ECs[8]. The epidermal growth factor receptor (EGFR) signalling pathway is also essential to promote ISC proliferation under both normal and stress conditions. Three *EGFR* ligands, *Spitz* (*Spi*), *Keren* (*Krn*) and *Vein* (*Vn*), as well as several Rhomboid-family intramembrane proteases that activate Spitz and Keren, are produced by ECs and visceral muscle under stress conditions. These factors trigger EGFR signalling in progenitor cells, promoting their growth and proliferation[9–12]. Inactivation of EGFR signalling in ISCs blocks their growth and division and impairs stem cell maintenance. So far, most studies of regenerative growth in the fly's midgut have focused on the mechanisms controlling ISC division, and little attention has been paid to the massive postmitotic growth that occurs in EBs as they endoreduplicate their DNA and differentiate into ECs.

The endocycle is an alternative cell cycle in which chromosome duplication occurs without nuclear division, resulting in polyploid cells that can be very large. Endocycles occur in most differentiated plant and invertebrate cells, as well as several vertebrate cell types[13]. ECs in the fly midgut undergo endoreplication as they differentiate, achieving maximum ploidies of about 64C (C value indicates chromatin copy number per cell, such that gametes are 1C and diploid cells are 2C or 4C). Although mitotic cycles and endocycles are distinct, they share many of the same regulatory components and mechanisms[14,15]. Both cycle types rely on a pulse of G1 cyclin/cyclin-dependent kinase (Cyc/Cdk) activity to trigger S phase, and

upstream transcriptional control of G1 cyclins by E2F/RB complexes is a common feature. In *Drosophila*, the periodic destruction of *E2f1* by the CRL4[CDT2] ubiquitin ligase is essential for endocycle progression, because this periodically quenches the expression of *CycE/Cdk2* and thereby allows the formation of pre-replication complexes on the DNA[16]. Suppression of mitotic genes such as *CycA* or *Cdk1*, or upregulation of APC/C[fzr], can convert mitotic cycles into endocycles and result in polyploid cells[17,18].

In *Drosophila* ovarian follicle cells, *Notch* signalling promotes the mitotic-to-endocyle switch by causing the downregulation of the *Cdk1* activator *String* (*Cdc25*) and the upregulation of *Fzr* (*Cdh1*)[19,20]. Similarly, *Notch* signalling is required for EB-to-EC differentiation and endoreplication in the fly's midgut. *Notch*-mutant ISCs remain diploid and mitotic, whereas hyperactive *Notch* signalling drives the switch to postmitotic endocycles, suggesting a similar mechanism as in the ovary. In many of *Drosophila* larval cells, reduction of nutrient-dependent InR/Pi3K/TOR (Insulin receptor/phosphoinoside 3 kinase/target of rapamycin) signalling inhibits the endocycle and results in small cells, whereas activation of Pi3K or TOR promotes cell growth and endocycling even under starvation conditions that normally cause arrest[21–24]. Zielke *et al.*[16] proposed that TOR-mediated growth promotes endocycling by upregulating the translation of *E2f1*, which in turn promotes transcription of genes required for DNA replication, most critically *cyclin E*. Similar effects have been reported in the adult midgut: for instance null mutations of *InR* can block EC endocycles, whereas artificially activating *InR* promotes increased EC growth[25,26].

Tissue size is determined by both cell size and cell number[27–30]. Most differentiated *Drosophila* larval cells become polyploid, and growth in most of the larva's tissues is driven primarily by increases in cell size rather than cell number. Analysis of the mechanisms of growth control in endocycling cells revealed that these cells respond to the same regulators of growth as diploid cells[31,32]. Recent work with the *Drosophila* ovarian follicular epithelium demonstrated that InR/Pi3K signalling controlled sporadic compensatory cellular hypertrophy by accelerating the endocycle, thus enhancing tissue repair after cell loss[33]. Another recent report documents induced endocycling and cell fusion as mechanisms of damage response in *Drosophila* adult abdominal epidermis, a tissue that lacks resident stem cells[34]. Apart from these two examples in flies and several interesting studies in the mammalian liver and cardiac muscle[13], cell growth driven by polyploidy has not been well investigated in the context of tissue homeostasis[13,35,36].

The study we present here details how EC growth mediated by endocycling is utilized by the fly midgut during damage repair. We find that the postmitotic growth of ECs is dependent upon endocycling and is essential for gut homeostasis and effective regeneration. In healthy flies, Insulin/Pi3K/TOR signalling promotes postmitotic EB/EC growth, but after gut epithelial damage EGFR/Ras/mitogen-activated protein kinase (MAPK) signalling drives postmitotic growth via a novel InR/Pi3K/TOR-independent mechanism. We furthermore find that the E2f1 transcription factor is required and sufficient to drive EB/EC endocycles, and that E2f1 is posttranscriptionally induced by Ras/MAPK signalling. Our study illustrates how distinct signalling pathways direct stress-dependent versus homeostatic regeneration, and highlight the importance of postmitotic cell growth and endoreplication in gut epithelial repair.

## Results

**Gut epithelial stress induces compensatory endoreplication.** The enteropathogen *Pseudomonas entomophila* (*P.e.*) secretes a virulence factor, haemolysin, that lyses ECs in the gut

epithelium[37]. Previous studies reported that gut damage from *P.e.* upregulates *EGFR* ligands (*Spi*, *Krn*, *Vn*) and ligand-activating proteases (Rho1, 2, 4, 6) to activate the EGFR/MAP Kinase pathway in ISCs, promoting ISC growth and division[10]. When we used *P.e.* to induce tissue damage we observed increased phospho-histone H3 (PH3)-positive cells and upregulation of reporters of *EGFR* ligand expression, as previously reported[8]. In addition, we detected higher ploidy in ECs than in control ECs from mock-infected animals, as assayed by both fluorescence-activated cell sorting (FACS) and quantitative imaging (Fig. 1a–d). As polyploidization often coincides with increased cell size, this extra endoreplication could result in larger ECs in the midgut[38]. This was confirmed by cell volume estimates derived from FACS forward scatter measurements following expression of activated Ras or Raf in EBs (Supplementary Fig. 1e). This suggested that after midgut injury, in addition to increased stem cell divisions, enhanced postmitotic EC growth may help to replace the mass of lost cells.

To test this hypothesis we assayed EC ploidy and size in midguts in which the generation of newborn ECs had been blocked by depleting *Notch* using RNA interference (RNAi) in stem cells[6]. This treatment also leads to the rapid overgrowth of ISC-like cells that displace ECs, causing their extrusion, and thereby accelerates EC attrition[39]. After 3 days, we observed only 20–40 ECs remaining in the midgut epithelium, but the nuclear DNA content of these cells was ∼5 times greater than normal ECs, indicating that they had undergone 2–3 extra endocycles (Supplementary Fig. 1a,b). These results demonstrate that extra cell growth and increased ploidy (cellular hypertrophy) occurs in response to different stress conditions.

To identify the origin of this higher ploidy we used the *esg-Gal4 UAS-Flp Act > CD2 > Gal4 UAS–GFP* system, hereafter referred to as '*escargot Flip Out*' (*esgF/O*)[8]. This heritable green fluorescent protein (GFP) marker system allows the researcher to trace the lineages of all ISCs in the regenerating midgut. Flies were shifted to 29 °C for 24 h, causing all progenitor cells (ISCs and EBs) to express GFP. We then fed the flies *P.e.* and 5-ethynyl-2'-deoxyuridine (EdU) for 8 h. EdU immunofluorescence revealed that > 90% of the large EdU-positive cells were also GFP positive. These cells were thus newborn ECs, as they inherited GFP expression from progenitor cells that had differentiated after the temperature shift. Only ∼5% EdU + cells were GFP negative (Fig. 1e,f). These results indicate that fully mature ECs do not generally re-enter the endocycle, even in response to stress-induced signalling that drives regeneration.

To determine whether EC endoreplication contributes to stress resistance we performed survival assays in which endoreplication had been blocked specifically in postmitotic enteroblasts by expression of DNA replication inhibitory factors. We used a conditional EB-specific driver, *Su(H)GBE-Gal4 tubGal80^ts*, to overexpress either *Geminin*, which binds and inhibits the DNA replication factor *Cdt1*, or an RNAi targeting DNA polymerase-α. Both transgenes effectively suppressed endoreplication in EBs and young ECs, resulting in smaller ECs (see, for example, Fig. 2e). Under normal culture conditions suppression of endoreplication in this way slightly reduced the flies' lifespan. However, following *P.e.* infection, the inhibition of endoreplication in EBs reduced survival time by 50% (Fig. 1g). Thus, cell growth driven by polyploidization is an essential aspect of the stress response.

## EGFR/MAPK is active in EBs and newborn but not mature ECs.
EGFR/MAP kinase signalling is induced by gut epithelial damage, and is essential for ISC proliferation during regeneration[10]. Thus, we hypothesized that *EGFR/MAPK* might also promote the growth and endoreplication of EBs and newborn ECs. To test

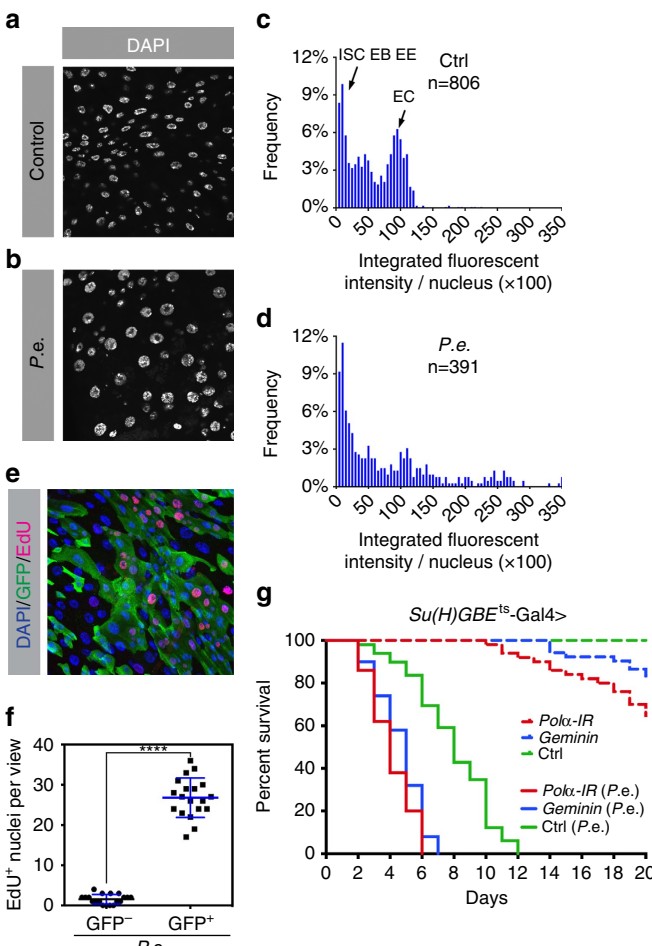

**Figure 1 | Epithelial damage induces extra endoreplication in ECs.** (**a–d**) Midguts of 3–5-day-old flies were stained with DAPI (blue) to visualize DNA. Representative images of (**a**) uninfected controls and (**b**) individuals that were infected with *P.e.* for 1 day. The majority of nuclei present in the midgut epithelium are EC nuclei. Quantification of EC nuclear DNA content (**c,d**) measured by integrated fluorescent intensity per nucleus from 10 midguts per genotype. *P.e.* infection led to higher EC ploidies. EE, enteroendocrine cell. (**e,f**) The *esgF/O* flies were shifted from 18 to 29 °C for 24 h. The *esg-Gal4* drove expression of *UAS-GFP* and *UAS-flp* that in turn led to *act-Gal4* and GFP expression in all *esg +* cells and their newborn progeny. After 24 h, these flies were fed *P.e.* and EdU in sucrose solution for 8 h. (**e**) Midguts were labelled with DAPI to visualize DNA (blue) and incorporated EdU (red) to identify endocycling cells. All *esg +* cells and their daughters produced during the experiment expressed GFP (green). (**f**) Quantification of EdU incorporation for 15 guts. Thus, 95% of EdU + cells were GFP +, while ∼5% of EdU + cells did not express GFP. ****P value from Student's *t*-test (P < 0.0001); n = 10. All experiments were repeated twice. (**g**) Survival curve showing that blocking endoreplication in EBs and pre-ECs reduces lifespan. *Geminin* or *DNA polymerase-α* (*Pol α* RNAi (inverted repeat (IR)) was overexpressed in EBs and pre-ECs using *Su(H)GBE-Gal4*. Dashed lines represent control flies raised in normal food with sucrose; solid lines represent experimental flies raised in food with *P.e.* bacteria.

this, we induced midgut stress in *esgF/O* flies by feeding *P.e.*, and then performed immunofluorescence to assess the expression active MAPK (di-phospho-ERK; dp*ERK*) and *EGFR* protein. Consistent with previous reports[9,11,12] dp*ERK* signals were confined to a subset of *esg +* progenitor cells in noninfected controls. After infection, however, newborn ECs present in GFP + *esgF/O* clones were also dp*ERK* positive (Fig. 2a).

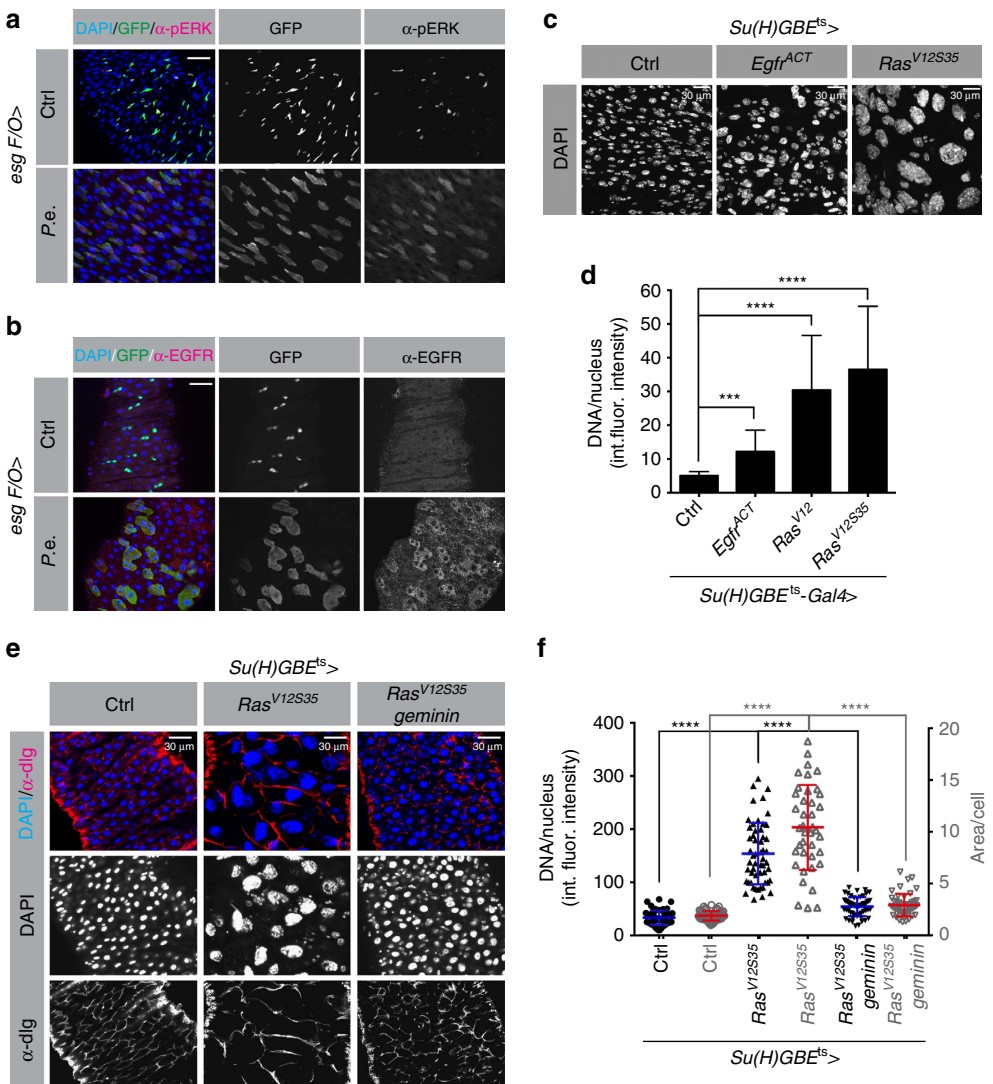

**Figure 2 | *EGFR* expression and function in enterocytes.** (**a,b**) The *esgF/O* system was used to mark *esg*+ cells and their newborn progeny generated during 24 h at 29 °C. Flies were then fed *P.e.* for 8 h. Midguts were then fixed and labelled with DAPI to visualize DNA (blue) and anti-*diphospho-ERK* (α-*pERK*, red (**a**)) or anti-*EGFR* antibodies (red (**b**)). (**a**) *P.e.* stress induced large clones (green) with increased *pERK* levels in newborn ECs (red). (**b**) *EGFR* expression was also higher in newborn GFP+ cells, including ECs, than in older GFP-surrounding cells. (**c,d**) *Su(H)GBE-Gal4ts*-driven overexpression of *RasV12*, *RasV12S35* or *EGFR*ACT in EBs and pre-ECs for 3 days caused hyperpolyploidy. In (**d**), each bar represents the average integrated DAPI intensity/ nucleus from 100 nuclei, from 10 midguts total. Error bars represent s.d. Student's *t*-test was used to determine statistical significance (***$P = 0.0001$ and ****$P < 0.0001$); int. fluor. intensity, integrated fluorescent intensity. (**e,f**) *RasV12S35* or *RasV12S35* and *Geminin* were overexpressed in EBs for 3 days. DNA content/nucleus was determined by DAPI signal as above (blue in (**e**) and black in (**f**)). Cell areas were measured using the cortical stain, anti-dlg antibody (red in (**e**) and grey in (**f**)). A total of 80 nuclei from 10 midguts total were counted. Larger nuclei correlated well with larger cell size. ****$P < 0.0001$ from Student's *t*-test.

Consistent with this, the expression of the *EGFR* protein was also much higher in newborn GFP+ ECs than in older GFP- ECs that existed before the infection (Fig. 2b). These data suggest that both the expression and activity of *EGFR* are upregulated in the newborn EBs and ECs that undergo endoreplication after infection stress.

To further investigate where and when EGFR/MAPK signalling is important in the midgut, we analysed the mRNA levels of EGFR/MAPK pathway components in the different midgut cell types from RNA-sequencing data[40]. The levels of the *EGFR*, *Ras*, *Raf* and *ERK* mRNAs were almost 10-fold higher in progenitor cells (ISCs and EBs) than in differentiated ECs (Supplementary Fig. 2). However, 48 h after *P.e.* infection, *EGFR* mRNA expression in ECs was increased ~20-fold[40]. Combined with our previous data, these results indicate that the EGFR/MAPK pathway is transcriptionally silenced during EC maturation. The high levels of dp*ERK* and *EGFR* mRNA and protein observed in newborn ECs (Fig. 2a,b) may reflect the inheritance of these factors from activated ISCs and EBs. This may explain the recalcitrance of mature ECs to *EGFR* ligands[10], and their inability to resume cell growth and endoreplication following infection.

**EGFR/MAPK signalling drives midgut EC endoreplication.** Next we asked whether activation of Ras/MAPK signalling was sufficient to drive endoreplication. We overexpressed activated forms of *Ras* or *Raf* in ECs using the *MyoIA-Gal4* driver for 1 day, dissected the intestines and incubated them *in vitro* for 3 h with EdU, a thymidine analogue that is incorporated into DNA during DNA synthesis. After 24 h of expression of activated *Ras* or *Raf*,

>90% of ECs were EdU positive, whereas wild-type controls showed only 4–5 EdU-positive ECs per midgut (Supplementary Fig. 3a). FACS analysis revealed that midguts expressing $Ras^{V12}$ contained a greater proportion of higher ploidy cells than controls (Supplementary Fig. 3b). We also noticed that many progenitor cells incorporated EdU when $Ras^{V12}$ was over-expressed in ECs. We believe this non-cell-autonomous effect is due to stressed ECs triggering ISC proliferation, a well-documented phenomenon[8]. These results indicate that MAPK signalling activity can activate EC endoreplication.

Although we detected an increase in ECs with higher ploidy in these experiments, the effect was small (Supplementary Fig. 3b). One possible explanation for this is that since $Myo1A$-$Gal4$, the driver used in these experiments, is active only in mature ECs, these cells were already somewhat refractory to $Ras$- or $Raf$-induced endocycle entry. To test this we did experiments with the $Su(H)GBE$-$Gal4$ driver that is expressed specifically in EBs and early-stage, immature ECs. In these samples we observed large increases in cell size and ploidy (Fig. 2c). Quantification of integrated DNA fluorescence intensity per nucleus showed that expression of $Ras^{V12}$ increased EC ploidy 8–10-fold (Fig. 2d). Overexpressing $Ras^{V12}$ using $Su(H)GBE$-$Gal4$ also increased both PH3$^+$ GFP$^-$ cells and PH3$^+$ GFP$^+$ cells. Thus, enforced $Ras$ activity in EBs induced autonomous endoreplication and both cell autonomous and nonautonomous cell proliferation.

$Ras^{V12}$ has the capability to activate both $Raf$, which activates MEK and MAPK/ERK ($rolled$), and $Pi3K$, which stimulates Insulin/IGF downstream components including $AKT$ and $TOR$[41]. Therefore, we investigated which signal transduction pathway downstream of $Ras$ contributed to growth endoreplication in the midgut. Two $Ras$ isoforms, $Ras^{V12S35}$, which activates $Raf$ but is defective in $Pi3K$ activation, and $Ras^{V12G37}$, which can activate $Pi3K$ but is defective in $Raf$ activation, were overexpressed using $Su(H)GBE$-$Gal4$. Only $Ras^{V12S35}$ could drive EB growth and endoreplication, whereas $Ras^{V12G37}$ was inert (Fig. 2d and Supplementary Fig. 1c,d). Unfortunately, as overexpression of $Raf^{GOF}$ by $Su(H)GBE$-$Gal4^{ts}$ was lethal, we could not determine its function in this context. However, overexpression of an active form of the $EGFR$ using the $Su(H)GBE$-$Gal4$ driver induced a 2–3 × increase in EB/EC DNA content (Fig. 2c). These results suggest that activation of the Raf/MEK/MAPK branch of the EGFR/Ras pathway is responsible for endoreplication in response to tissue damage.

Higher ploidy correlates with larger cell size in most polyploid cells, and in some cases endoreplication has been shown to be required for cell size increases[14,42]. To determine whether this is the case in the midgut we measured the areas of $Ras^{V12S35}$ expressing cells using anti-Dlg immunofluorescence that highlights the cell cortex. We found that the increase in area/cell affected by $Ras^{V12S35}$ expression was roughly proportional to the increase in nuclear DNA, namely ~10-fold (Fig. 2e,f). To determine whether endoreplication was necessary for the increased size of these cells we blocked DNA replication by overexpressing $Geminin$, a specific inhibitor of DNA replication, at the same time that $Ras^{V12S35}$ was overexpressed. This effectively suppressed the increase in cell size (Fig. 2e), demonstrating that endoreplication is required for EC growth.

**EGFR/Ras activity requirements at homeostasis.** Upregulation of Ras/Raf signalling in EBs or ECs results in a hyper-endoreplication phenotype similar to that observed following regenerative growth in response to $P.e.$ infection. Stress from infection and other damaging agents strongly induces EGFR/MAPK signalling in this tissue, and the depletion of $EGFR$ and its downstream effectors can strongly suppress ISC proliferation in response to tissue damage[9,10].

However, whether loss of MAPK signalling suppresses cell growth in endocycling cells has not been determined. To test this, we first tried suppressing the MAPK pathway in ECs using the $MyoIA$-$Gal4$ driver to deplete $EGFR$ or $Ras$ with RNAi, or by overexpressing $Map Kinase Phosphatase 3$ ($MKP3$) or $Cbl$, a ubiquitin ligase that promotes $EGFR$ degradation. We did not observe significant reductions in cell size or ploidy following these manipulations. We suspected that this lack of effect might be due to the late activation of the $MyoIA$ driver during EC maturation. Thus, to suppress EGFR signalling earlier in EC development, we made MARCM (mosaic analysis with a repressible cell marker) clones with $EGFR$-null mutations in ISCs. As previously reported[10], at 5 days after clone induction, ISC mitoses in $EGFR^{-/-}$ clones were strikingly decreased. However, endoreplication still occurred in the few mutant ECs that were generated, and the DNA content of these ECs was not significantly reduced (Supplementary Fig. 4a–c). Next, we generated $Ras85D$-null mutant clones in the midgut. To be sure that the scored clone cells were ECs, we stained the samples with anti-$PDM1$ antibody, a marker of EC differentiation. Many midgut cells mutant for $Ras$ were positive for $PDM1$, demonstrating that $Ras$ is not required for EC differentiation (Supplementary Fig. 4f). Although most $Ras85D^{-/-}$ clones consisted only of single cells[10], DNA quantification showed that the amount of DNA/cell was only slightly less than in wild-type control ECs (Fig. 3a). These results are consistent with the previous outcome from knockdown of EGFR/Ras pathway components by $UAS$-$RNAi$, in which no significant effect on endoreplication was detected. We conclude that although EGFR/Ras activity is important for ISC mitoses, it may not be essential for EC endoreplication under normal homeostatic conditions.

**InR/Pi3K/TOR is required for EC growth at homeostasis.** InR/Pi3K/TOR signalling is limiting for nutrition-dependent cell growth and endoreplication in many $Drosophila$ organs[21,24,43]. Consistent with this, we found that overexpression of $InR$ or the $TOR$ activator $Rheb$ in EBs using the $Su(H)GBE$-$Gal4^{ts}$ driver produced larger cells with higher ploidy. $InR$ activity has also been reported to be required for EC differentiation[44]. To test the role of InR/Pi3K/TOR signalling in EC endoreplication, we generated ISC clones mutant for null alleles of the $InR$ ($InR^{339}$), $Pi3K$ ($Dp110^A$) or $TOR$ ($TOR^{ΔP}$), using the MARCM technique. At 5 days after clone induction, most $InR^{-/-}$-, $Pi3K^{-/-}$- or $TOR^{-/-}$-mutant clones were arrested as single cells, though a few clones had two or three cells. Most wild-type control clones, in contrast, had >3 cells, and 50% contained >8 cells (Figs 3d,g and 4a,d and Supplementary Fig. 4d). Thus, ISC division was strongly dampened by loss of $InR$, $Pi3K$ or $TOR$ function. Moreover, in agreement with previous reports[25,26], all of the $InR$-mutant cells were arrested in the diploid state. $Pi3K$- and $TOR$-mutant ISCs also arrested as diploids (Fig. 3d,g). Quantification of DNA content in these and control clones showed that wild-type ECs contained 5–8 times more DNA than $InR$, $Pi3K$ or $TOR$-mutant cells (Fig. 3f,i). Hence, InR/Pi3K/TOR signalling is essential for DNA endoreplication in EBs and ECs under normal physiological conditions.

**Ras/Raf drives stress-dependent EC growth without InR/TOR.** We next asked which of these genes were required for cell growth and endoreplication following gut stress. After inducing $EGFR$-, $Ras$-, $InR$-, $Dp110$- or $TOR$-mutant MARCM clones for 2 days, we initiated gut epithelial damage by feeding the flies $P.e.$ for 1 day to generate an infection. In wild-type controls, $P.e.$ stress generated larger ISC clones comprising ECs with larger nuclei and higher ploidy, as described above. However, with the same $P.e.$ stress, clones homozygous for $EGFR^{CO}$- or $Ras^{ΔC40B}$-null alleles

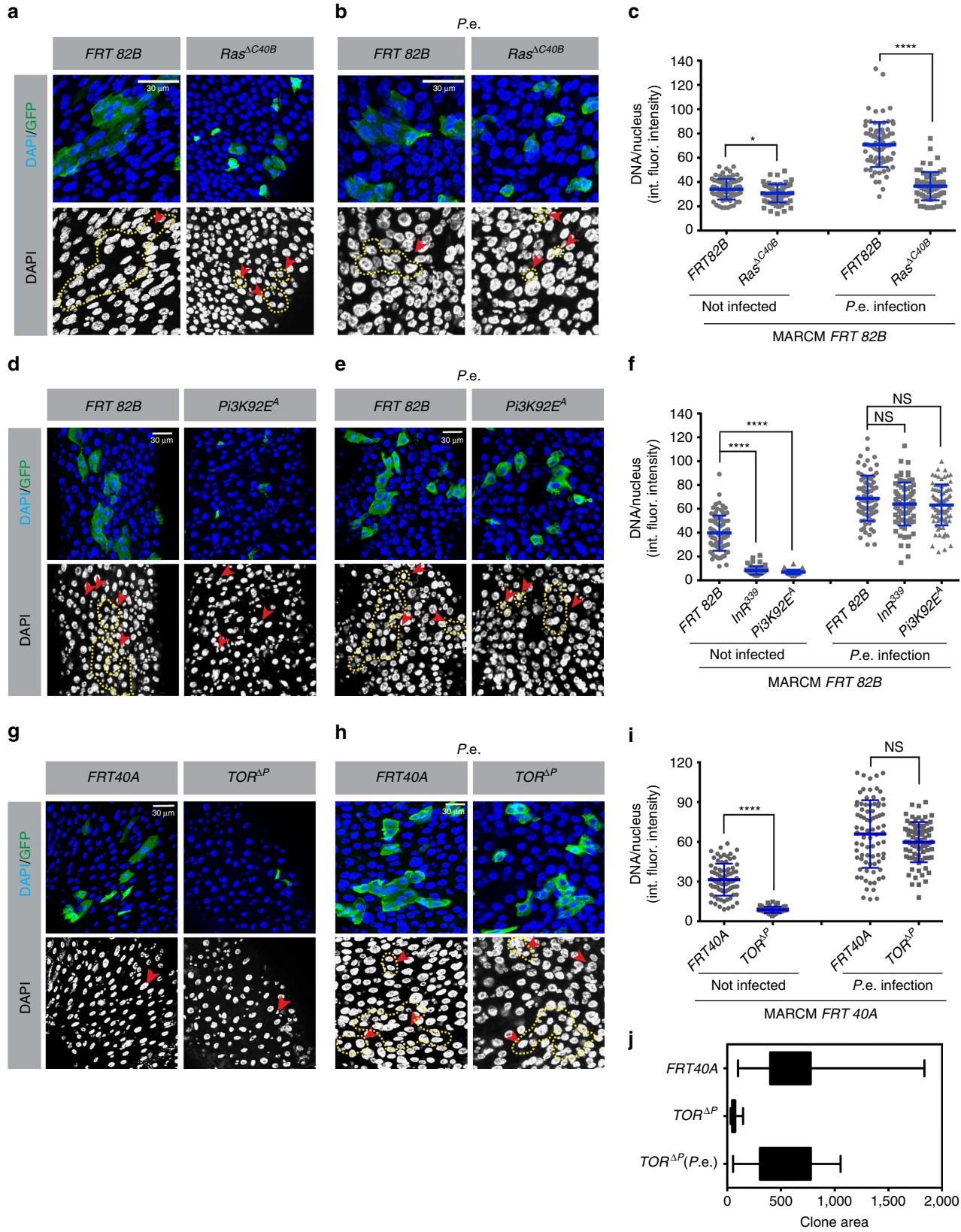

remained arrested as single cells that contained significantly less DNA than control cells (Fig. 3b and Supplementary Fig. 4b). These mutant cells, however, were not diploid, but merely underreplicated. We conclude that EGFR/Ras signalling is essential for stress-induced ISC divisions and clonal growth, and

is also important for normal levels of stress-induced EB/EC endoreplication.

We next tested the requirement for InR/Pi3K/TOR signalling under the same conditions. Bacterial infection was carried out as above, in flies bearing MARCM clones homozygous for null

alleles of *InR*, *Pi3K*, or *TOR*. In contrast to the results obtained in noninfected flies, these mutant clones contained many cells, and the ECs in these clones had high ploidies similar to those in ECs from stressed wild-type controls (Fig. 3e,h and Supplementary Fig. 4e). In the case of *TOR* mutants we also measured clone areas to gauge total amounts of growth in ISC lineages following infection. Following *P.e.* infection, *TOR*-mutant clones achieved sizes comparable to wild-type controls (Fig. 3j). These results indicate that following infection stress, InR/Pi3K/TOR signalling is not required for ISC division or for EB/EC growth and endoreplication, whereas EGFR/Ras/MAPK signalling is.

To further test the function of EGFR signalling in midgut cell growth and endoreplication we performed epistasis assays in which *InR*-, *Pi3K*- or *TOR*-null mutant clones were induced using the MARCM system, and either $Ras^{V12S35}$ or $Raf^{GOF}$ was simultaneously overexpressed in the clones. As described above, *InR*-, *Pi3K*- or *TOR*-mutant clones arrested as one or two diploid cells. However, when either $Ras^{V12S35}$ or $Raf^{GOF}$ was overexpressed in *InR*-, *Pi3K*- or *TOR*-mutant ISCs, multiple-celled clones containing highly polyploid ECs grew out (Figs 4 and 5). To further exclude the importance of *TOR* in mediating *Ras/Raf*-driven growth, we fed flies the *TOR* inhibitor rapamycin for 2 days before clone induction. We then generated midgut cell clones that were homozygous for a *TOR*-null allele and also overexpressed $Ras^{V12S35}$. This treatment was designed to inhibit any residual *TOR* protein that remained after the *TOR* gene was deleted. At 5 days after clone induction and rapamycin treatment, *TOR*-mutant clones expressing $Ras^{V12S35}$ contained multiple polyploid ECs, whereas control clones mutant for *TOR* but not expressing $Ras^{V12S35}$ comprised mostly small single-arrested cells, presumably in the diploid state (Supplementary Fig. 5). Altogether, these results indicate that enforced Ras/Raf signalling can drive ISC mitoses and EB/EC growth and endoreplication in the absence of *InR*, *Pi3K* or *TOR* activity. Considering that *P.e.* infection was similarly able to bypass the requirements for *InR*, *Pi3K* and *TOR* for growth, and that Ras/Raf signalling is strongly induced by *P.e.* infection, these data support the conclusion that EGFR/Ras/MAPK signalling is necessary and sufficient to drive both ISC and EB/EC growth during stress-induced midgut regeneration. In contrast, InR/Pi3K/TOR appears to be dispensable during stress-induced gut epithelial renewal, but essential for growth in nonstressed conditions.

The *Myc* transcription factor is known as a potent regulator of metabolism, protein synthesis and cell growth. *Myc* can be upregulated by Ras signalling in both human and *Drosophila* cells[41], and it is induced and required for midgut regeneration[45]. In certain types of *Drosophila* cells, enforced *Myc* expression is sufficient to drive not only growth but also endoreplication[46]. To test whether *Drosophila Myc* might be sufficient to mediate the effects of Ras/Raf signalling, we overexpressed it in EBs for 3 days using the $Su(H)GBE$-$Gal4^{ts}$ driver. Contrary to our expectations, increased cell size and higher ploidy were not observed in these cells. To test whether *Myc* was required for EC endoreplication, we generated $Myc^3$ null mutant clones using the MARCM technique. Consistent with RNAi results reported by Ren *et al.*[45], $Myc^3$ mutant cells arrested at the diploid stage with small size (Supplementary Fig. 6b). Thus, *Myc* is required for EB/EC growth and endoreplication, but not sufficient to induce endoreplication or growth in these cells.

**Ras/Raf posttranscriptionally upregulates *E2f1*.** The conserved transcription factor, *E2f1*, controls expression of many cell cycle genes and is essential for endocycle progression in various species[13]. Our studies in *Drosophila* salivary glands demonstrated that endocycle progression is directed by *E2f1* protein oscillations that result from the periodic degradation of *E2f1* during S phases by the replication fork-dependent ubiquitin ligase, $CRL4^{Cdt2}$. Furthermore, this study suggested that modulation of *E2f1* translation in response to TOR signalling affects endocycle rates and final ploidy[16]. Therefore, we tested the function of *E2f1* as a limiting regulator of EB/EC endocycles. First, we overexpressed *E2f1* or *E2f1* with its dimerization partner, *Dp*, in ECs using the *MyoIA-Gal4* driver. EdU labelling showed that this treatment activated ectopic DNA replication, and DNA measurements showed that ECs had progressed through 2–3 extra endocycles after a 3-day induction of *E2f1* (Fig. 6a–c). To test whether *E2f1* is required for EC endoreplication we generated cell clones homozygous for $E2f1^{7172}$, a null allele. At 7 days after induction, these clones consisted of 1–2 diploid cells that had not endoreplicated (Fig. 6d). Importantly, some of these small cells were positive for the EC differentiation marker, *Pdm1* (Supplementary Fig. 6a), indicating that *E2f1* is not necessary for EC differentiation. These observations demonstrate that *E2f1* is essential for both ISC division and EB/EC endoreplication, and limiting for endoreplication in ECs.

Based on the above data we considered a model in which the increase in EGFR/Ras/MAPK signalling that occurs during midgut regeneration upregulates *E2f1* that in turn promotes endoreplication. To explore this possibility we employed a reporter of *E2f1* transcriptional activity, *PCNA*-GFP[47]. *PCNA* is a cofactor of DNA polymerase Delta that increases the processivity of leading strand synthesis during DNA replication. The *PCNA* promoter present in the *PCNA-GFP* reporter contains three transcriptional regulatory elements: a URE (upstream regulatory element), a DREF recognition site (DNA replication-related element) and two *E2f* recognition sites. The role of *E2f* in controlling *PCNA* expression during the cell cycle is well documented and *PCNA* expression is commonly used as a reporter for *E2f1* activity[48]. To test the effect of Ras/Raf signalling

**Figure 3 | Clonal growth and EC ploidy affected by *Ras Pi3K TOR* and *P.e.* infection.** $Ras85D^{\Delta C40B}$-, $Pi3K92E^A$- or $TOR^{\Delta P}$-null mutant clones were made using the MARCM system. At 5 days after clone induction, midguts were dissected. Integrated DAPI intensity/nucleus was measured to determine ploidy. Clones were marked with GFP (green) and stained for DNA (blue). Clone boundaries are outlined by yellow dashed lines. (**a**) Flies raised on normal food. Left panel indicates FRT control and right panel *Ras85D*-mutant clones. (**b**) At 2 days after clone induction, flies were orally infected with *P.e.* for 24 h. Left panel indicates FRT control and right panel *Ras85D*-null mutant clones. (**c**) Quantification of DNA content/nucleus in clonally marked cells for (**a,b**). A total of 100 GFP⁺ nuclei from 10 midguts total were scored for each genotype. *Ras85D*-mutant cells had a slightly lower ploidy than wild-type control cells in normal conditions. *Ras85D*-mutant cells could not divide or become hyperpolyploid following stimulation by *P.e.* infection. int. fluor. intensity, integrated fluorescent intensity. (**d–f**) Experiments similar to those shown above, but using a null mutation in *Pi3K* or *InR*. A total of 80 GFP⁺ nuclei from 10 (control, InR mutant) or 20 (Pi3K mutant) midguts were scored. *Pi3K*- and *InR*-mutant cells had lower ploidy and reduced cell size under normal culture conditions, and appeared to be arrested in the diploid state. However, *P.e.* infection induced these mutant cells to divide and differentiate high ploidy ECs. Pictures of InR-mutant clones can be found in Supplementary Fig. 4 (**d,e**). (**g–j**) A similar experiment as above, but with *TOR*-mutant clones. (**j**) Quantification of clone areas (yellow dashed line). A total of 80 GFP⁺ nuclei from 10 (control) or 25 (TOR mutant) midguts were counted for each genotype. *TOR*-mutant cells arrested as diploids in normal culture (right panel (**g**)), but *P.e.* infection induced these mutant cells to proliferate and generate normally polyploid ECs (right panel (**h**)). Error bars represent s.d. Student's *t*-test was used to determine statistical significance (*$P = 0.0108$, ****$P < 0.0001$, NS $P > 0.05$). All experiments were repeated three times.

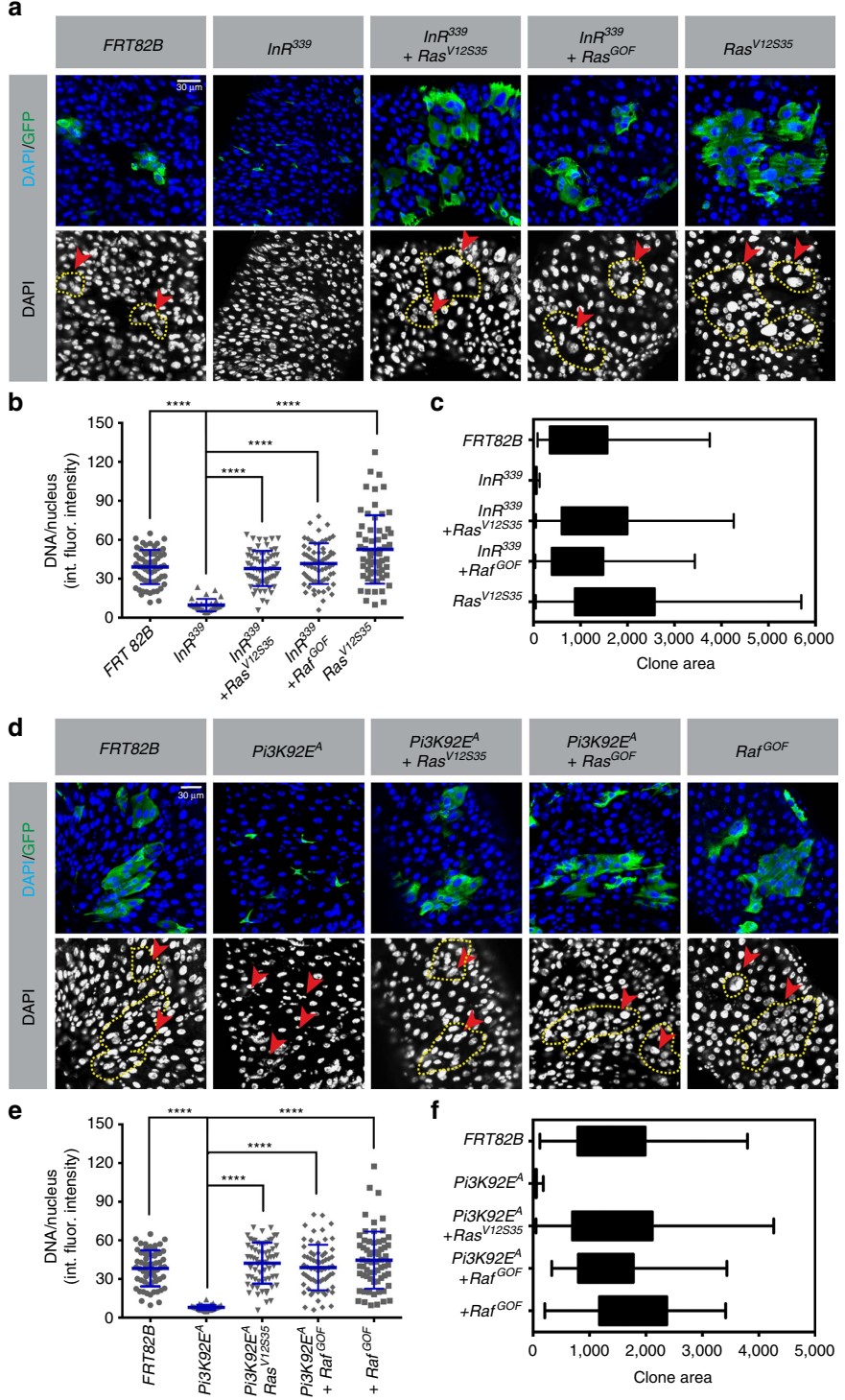

**Figure 4 | *Ras/Raf* signalling promotes EC endoreplication independently of *InR* and *Pi3K*.** *InR*- (**a**) or *Pi3K*- (**d**) null mutant cell clones were generated in the adult midgut using the MARCM system, as in Fig. 3. The MARCM system was also used to coincidently express $Ras^{V12S35}$ or $Raf^{GOF}$ in the mutant clones, as indicated. Midguts were dissected 5 days after clone induction. DNA was stained with DAPI (blue) and clones were marked with GFP (green) as outlined by yellow dashed lines. int. fluor. intensity, integrated fluorescent intensity. Nuclear DNA content (**b**,**e**) and clone areas (**c**,**f**) were measured as in Fig. 3. Control clones expressing either $Ras^{V12S35}$ or $Raf^{GOF}$ alone, from the same experiment. A total of 80 nuclei from GFP$^+$ cells from 10 total midguts were scored for each genotype, except for *InR*- and *Pi3K*-mutant clones, in which case 80 nuclei from 20 or 25 midguts were scored, respectively. Red arrowheads indicate representative GFP$^+$ nuclei. Error bars represent s.d. Student's $t$-test was used to determine statistical significance (****$P < 0.0001$). Expression of $Ras^{V12S35}$ or $Raf^{GOF}$ in $InR^{-/-}$- or $Pi3K^{-/-}$-mutant cells rescued the growth arrest of these cells, allowing them to divide and grow into multiple polyploid ECs. All experiments were repeated three times.

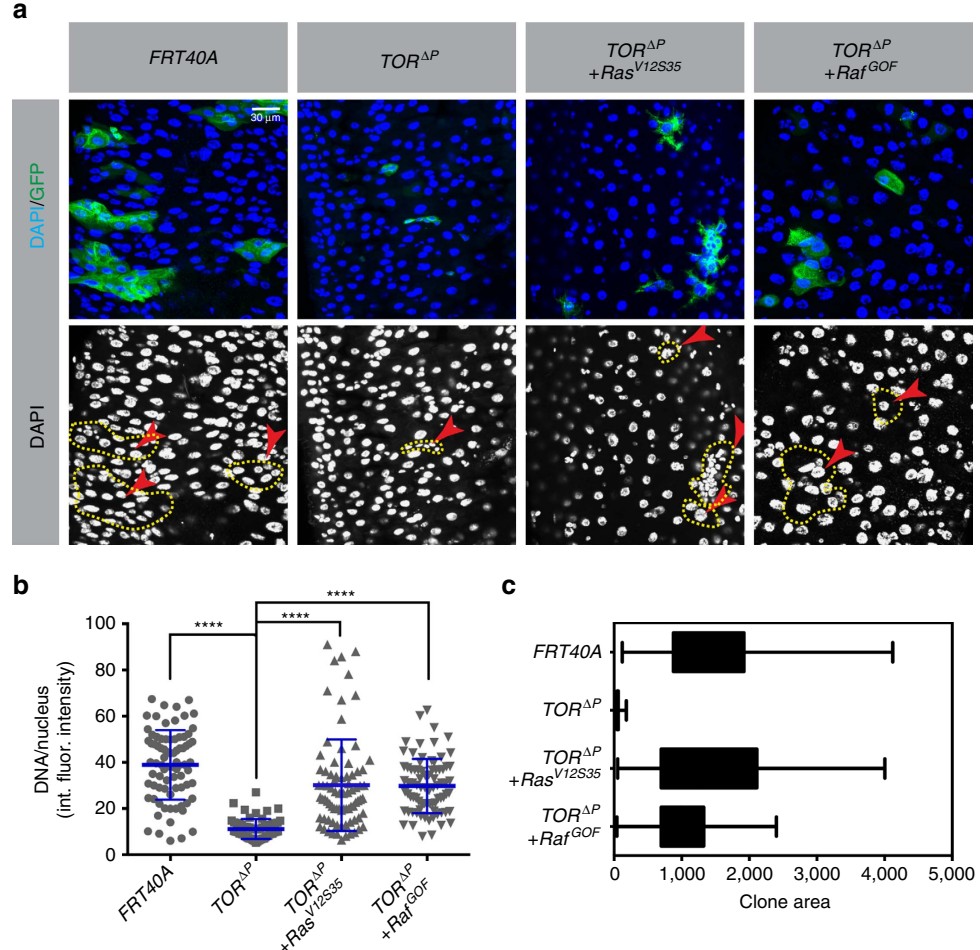

**Figure 5 | Ras/Raf signalling promotes EC endoreplication independently of TOR.** *TOR*-mutant cell clones were generated in the midgut using the MARCM system, as in Fig. 3. *Ras^V12S35* or *Raf^GOF* was coincidently expressed in the mutant clones, as indicated. Midguts were dissected 5 days after clone induction. (**a**) DNA was stained with DAPI (blue) and clones were marked with GFP (green) as outlined by yellow dashed lines. Nuclear DNA content (**b**) and clone areas (**c**) were measured as in Fig. 3. Error bars represent s.d. Student's *t*-test was used to determine statistical significance (****$P \leq 0.0001$). Expression of *Ras^V12S35* or *Raf^GOF* in *TOR^{-/-}*-mutant cells rescued the growth arrest of these cells, allowing them to divide and grow into multiple polyploid ECs. All experiments were repeated three times.

on E2f1 activity we overexpressed either UAS-*Ras^V12S35* or UAS-*Raf^GOF* in flies carrying the *PCNA*-GFP reporter using the ubiquitous inducible *Act Flip/Out Gal4* driver. At 3 days after induction, GFP was nearly undetectable in controls. In contrast, high levels of GFP were observed in midgut cells overexpressing either *Ras^V12S35* or *Raf^GOF* (Supplementary Fig. 7). This indicates that Ras/Raf/MAPK signalling upregulates *E2f1* activity. To confirm that the increased expression of PCNA-GFP was *E2f1* dependent, we used flies transgenic for *PCNA*-GFP^{ΔE2F1}, a variant reporter that carries mutations in the two *E2f1* recognition sites in the *PCNA* promoter. These mutations abrogate *E2f1* binding. In this case GFP was not detected in *Ras^V12S35*- or *Raf^GOF*-expressing midgut cells (Supplementary Fig. 7), supporting the conclusion that Ras/Raf/Mapk signalling upregulates *E2f1* activity.

Finally, we investigated the mechanism via which Ras/Raf activity upregulates *E2f1*. We first assayed levels of *E2f1* mRNA after stimulating Ras/Raf/MAPK signalling in ECs. Although the expression of three *E2f1* target mRNAs (*PCNA*, *RnrS* and *Cyclin E*) increased six–eight-fold when activated Ras or Raf was expressed in ECs, the mRNAs encoding *E2f1* and *E2f2* were not appreciably induced (Fig. 6f). Western blot analysis, however, showed that the amount of *E2f1* protein in the midgut was

increased 6–12-fold by enforced Ras/Raf/MAPK signalling (Fig. 6g and Supplementary Fig. 8). This indicates that the high levels of *E2f1* activity induced by Ras/Raf signalling are a result of increased synthesis or stability of *E2f1* protein, but not its transcriptional upregulation.

## Discussion

Recent studies have revealed much about the molecular mechanisms that generate polyploidy, but comparatively little regarding how polyploidization is used physiologically. In cardiac muscle and the liver, endoreplication and cellular hypertrophy are believed to be essential responses to maintain organ function under stress[13]. In *Drosophila* glia, polyploidization links tissue growth to blood–brain barrier function[49]. In many other instances in plants and invertebrates, polyploidy-mediated cell growth is an adaptation that allows nutrient-dependent increases in tissue size during development[13], without cell division. However, despite that many epithelia, including human skin, are known to incorporate polyploid cells, few reports address polyploidization as a mechanism of epithelial renewal. In this study we show that EC endoreplication and the accompanying cell enlargement are essential for rapid regeneration of the gut

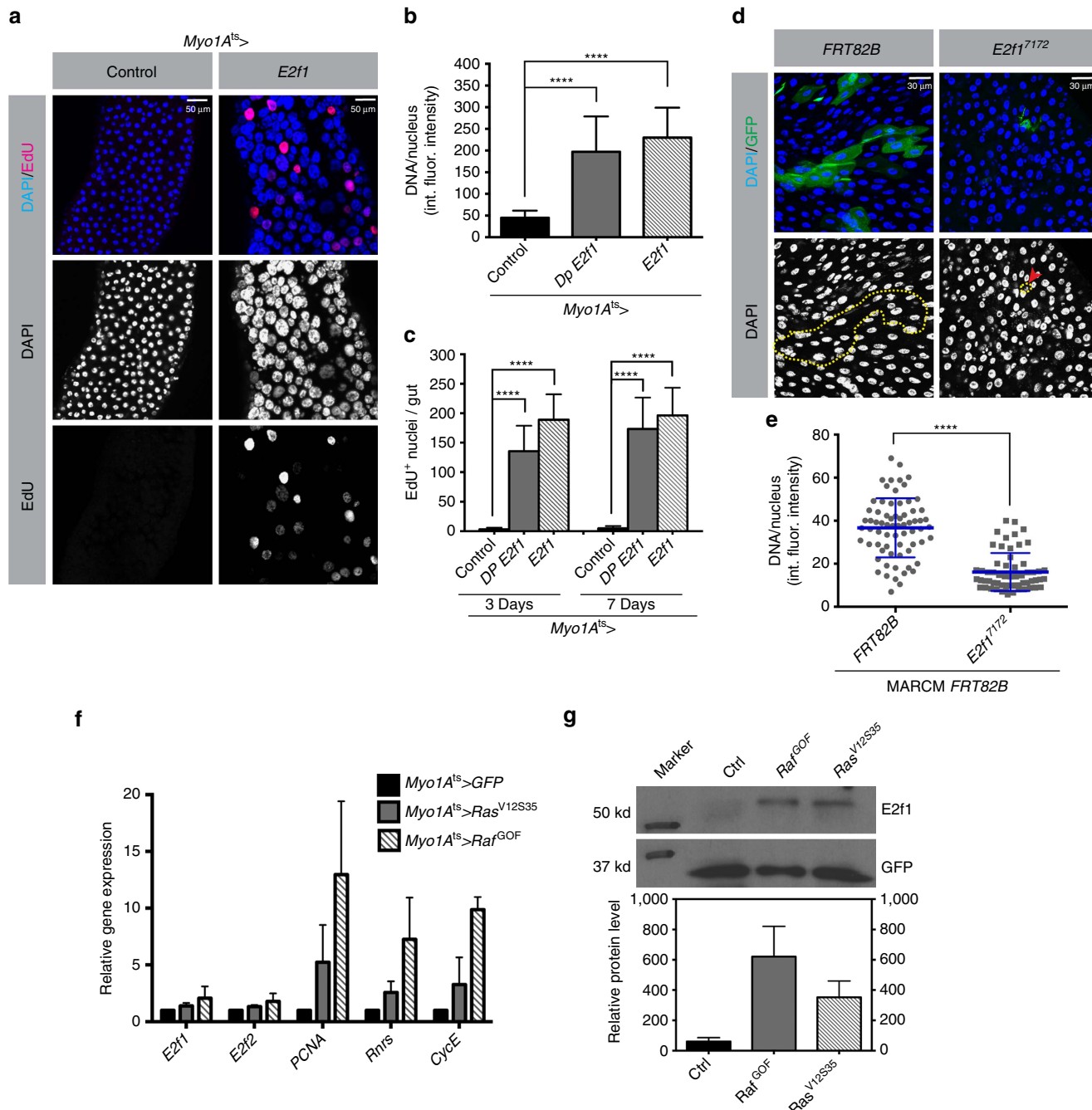

**Figure 6 | _E2f1_ controls endoreplication and is regulated post-transcriptionally by Ras/Raf signalling.** (**a–c**) _E2f1_ or _E2f1_ and _Dp_ were overexpressed in ECs using the _Myo1A-Gal4ts_ driver. After 3 days, midguts were dissected and incubated with EdU for 2 h, fixed and stained with DAPI to visualize DNA (blue), and for incorporated EdU (red) to visualize replicating DNA. Quantification of DNA content/nucleus using integrated fluorescence (int. flour.) intensity measurements (**b**) and of DNA replication using EdU incorporation (**c**) showed that overexpressed _E2f1_ promoted EC endoreplication and increased ploidy. A total of 100 nuclei from 10 total midguts for each genotype were quantified. (**d,e**) GFP-marked _E2f1_[7172]-null mutant cell clones were generated using the MARCM technique. At 7 days after clone induction, midguts were fixed and stained with DAPI. _FRT82B_ indicates control clones. (**e**) Quantification of nuclear DNA contents for wild-type and _E2f1_-mutant clone cells. A total of 80 GFP[+] nuclei from 10 control midguts or 20 _E2f1_ mutant midguts were scored. Loss of _E2f1_ blocked endoreplication, giving lower DNA contents/nucleus. (**f**) Reverse transcription quantitative PCR (RT-qPCR) was used to measure the mRNA levels of the indicated genes, following 3 days of overexpression of _Ras_[V12S35] or _Raf_[GOF] in ECs using the _Myo1A-Gal4ts_ driver. A total of 15 midguts for each genotype were used. RT-qPCR was used to measure the transcripts levels of _E2f1_, _E2f2_, _PCNA_, _Rnrs_ and _CycE_. Ras/Raf signalling in ECs increased the expression of the _E2f1_ target genes _PCNA_, _RnrS_ and _CycE_, but did not increase the levels of the _E2f1_ and _E2f2_ mRNAs. (**g**) Midguts expressing GFP alone (control) or GFP and _Ras_[V12S35] or _Raf_[GOF] under the control of _Myo1A-Gal4ts_ were dissected after a 3-day induction, and then total protein was isolated. Levels of _E2f1_ and GFP protein were assayed by immunoblotting. Graph shows _E2f1_ levels normalized to GFP from the same samples. Error bars represent s.d. Student's _t_-test was used to determine statistical significance (****$P < 0.0001$). _E2f1_ protein was upregulated in midguts that overexpressed _Ras_[V12S35] or _Raf_[GOF]. All experiments were repeated three times.

epithelium, and for maintaining barrier function, just as is the production of new ECs by ISCs.

Under conditions of high demand for epithelial surface, such as after enteric infection or in the absence of competent ISCs, we also noted compensatory EC hypertrophy (that is, increases in cell size and ploidy above normal). Compensatory hypertrophy is a strategy used in a variety of organisms and cell types. Recently, Tamori and Deng[33] demonstrated that in the *Drosophila* ovarian follicular epithelium, local cell loss triggers surrounding cells to undergo compensatory hypertrophy, and recent studies of the fly's hindgut and abdominal epidermis indicate that these organs can also undergo inducible repair involving dramatic increases in cell size and ploidy[34,50]. Similarly, in the mammalian liver, hepatocyte polyploidy-driven hypertrophy contributes to damage repair and recovery[51]. In the *Drosophila* midgut, we found that suppressing endoreplication in EBs and pre-ECs reduced the survival of flies following damage from bacterial infection. While this suggests that the capability for compensatory hypertrophy may also enhance fitness during gut stress, our experimental design made it impossible to distinguish the requirement for normal endoreplication from that for extra endoreplication and hypertrophy, and hence the necessity for hypertrophy in this context remains a matter of conjecture.

Homeostatic mechanisms control midgut cell number and size to preserve organ integrity and function. These mechanisms rely on a balance of positive and negative regulatory stimuli that maintain the tissue, altering cell growth rates to adapt to environmental conditions. Previous work demonstrated that during gut epithelial repair, EGFR/MAPK signalling is induced and stimulates ISC proliferation. We show here that damage-dependent MAPK activity also stimulates the growth of postmitotic enteroblasts and newborn ECs, where it is required and sufficient to promote endocycling. Interestingly, mature, fully differentiated ECs did not show evidence of MAPK activity or *EGFR* expression, and could not efficiently re-enter the endocycle in response to stress signalling. In these mature ECs, core components of MAP kinase pathway, including the *EGFR*, were transcriptionally downregulated. The loss of these factors and consequent insensitivity of fully differentiated ECs to growth factor signalling may be an important fail-safe mechanism that prevents excessive overgrowth after stress.

Our previous studies using larval salivary glands demonstrated that endocycles in that organ are driven by the *E2f1*, and that *E2f1* levels and activity were determined posttranscriptionally by nutrition-dependent *TOR* signalling[16]. We proposed that *TOR*-dependent translational control of *E2f1* determined endocycle rates. In this study we show that *E2f1* activity is also rate-limiting for polyploidization in midgut enteroblasts and enterocytes. As in salivary glands, *E2f1* levels in the midgut are posttranscriptionally increased by growth signalling, but in this case the critical input, at least after gut stress, is Ras/MAPK signalling. The effect of Ras/Raf signalling on *E2f1* levels could be due either to increased translation or to increased stability during Gap phases or both. While MAPK-dependent stabilization of *E2f1* is plausible, we favour a translational mechanism. This is because *E2f1* is degraded during each S phase via the S-phase-dependent CRL4$^{Cdt2}$ ubiquitin ligase, and hence its half-life is expected to be short in all cycling cells[16,52]. Indeed, the half-life of *E2f1* is expected to decrease, not increase, as the cell cycle accelerates. A MAPK-dependent increase in *E2f1* stability would therefore have to be mediated by an unknown Gap-phase-specific degradation pathway. As discussed below, how Ras/MAPK might increase the translation of *E2f1* remains an open question.

Currently, the best-characterized signalling system that regulates cell growth is the InR/Pi3K/Akt/TOR network. This system is important for controlling cell growth and body size

in most if not all animals, and its activity has striking effects on biomass and DNA content in many polyploid cells[21,22,24,53,54]. As expected, we found that $InR^{-/-}$-, $Pi3K^{-/-}$- or $TOR^{-/-}$-null mutant ISCs did not divide appreciably and could not produce large polyploid ECs. When enteric bacterial infection was used to damage the midgut, however, we were surprised to find that $InR^{-/-}$-, $Pi3K^{-/-}$- or $TOR^{-/-}$-mutant ISCs divided many times and generated ECs that grew to normal sizes through polyploidization. Based on this observation we propose that while the InR/Pi3K/TOR pathway is required for EC growth under normal physiological conditions, perhaps in response to nutritional inputs, it is dispensable during damage-induced regeneration. Further experiments indicated that under stress conditions, the MAP kinase branch of the *EGFR* signalling pathway drives midgut cell growth and endoreplication independently of InR/Pi3K/TOR signalling (Fig. 7). In this respect, the mechanism that promotes compensatory hypertrophy in the midgut may be distinct from that in *Drosophila* ovarian follicle, in which InR/Pi3K signalling appears to be important[33]. There are few reports of *Pi3K*-independent, *TOR*-independent cell growth in the literature, and the underlying mechanism for this is unknown.

In searching for this mechanism we considered the possibility that the ribosomal S6 kinase (*S6K*), a translational regulator that is activated by TOR, might also be a critical effector of Ras/Raf signalling in the fly midgut. In other *Drosophila* cell types, *S6K* can modulate cell growth[28], and in one intriguing case (larval neuroblasts) this appeared to be an *InR*- and *TOR*-independent activity[55]. However, as previously reported[56], we found that $S6K^{I-1}$-null mutant ISC clones grew normally and generated normally sized, polyploid ECs. Thus, *S6K* is not likely to be a key effector of EGFR/Ras/MAPK signalling during midgut regeneration.

Another potential effector of Ras-driven cell growth and endoreplication is the *Myc* transcription factor. *Myc* is responsive to increased *Ras* activity in both human and *Drosophila* cells[41], and is a potent driver of ribosome biogenesis, protein synthesis and cell growth in both contexts[46,57]. Moreover, *Myc* is important for ISC growth in the *Drosophila* midgut and also promotes Ras-mediated tumorigenesis there[45,58]. In testing the role of *Myc*, we found that while it is essential for ISC division and growth, it was not sufficient to promote either ISC division or EB/EC growth and endoreplication when overexpressed. Thus, while *Myc* might contribute to *Ras*-mediated cell growth in the gut, additional essential gene functions are targeted.

We recently reported that the transcriptional repressor *Capicua* and the transcriptional activators *Pointed* and *Ets21C* are important MAPK effectors in the *Drosophila* intestine, where they regulate ISC proliferation in response to infection stress[59]. Could these transcription factors also mediate Ras/Raf signalling in postmitotic EBs and ECs? Surprisingly, our tests showed that loss of *Capicua* or overexpression of *Pointed* or *Ets21C* specifically in enteroblasts did not generate large cell/hyperpolyploid phenotypes like those seen in response to activated *EGFR*, *Ras* or *Raf* signalling. Thus, our results leave it unclear how Ras/Raf/MAPK signalling promotes EB growth and endoreplication. We suggest that cytoplasmic effects on translation via novel effectors of MAPK are a likely mechanism. Future studies of how Ras/MAPK signalling promotes *TOR*-independent cell growth should prove to be very interesting, and promise insight into how to treat the many cancers in which Ras/Raf signalling components are mutationally activated.

## Methods

**Fly stocks and genetics.** UAS transgenes are as follows: UAS-Ras$^{V12}$, UAS-Ras$^{V12S35}$, UAS-Ras$^{V12G37}$, UAS-Raf$^{GOF}$, UAS-MPK3, UAS-EGFR(WT), UAS-λTOP, UAS-N$^{intra}$ and UAS-spi.s.4A. RNAi transgenes used are as follows:

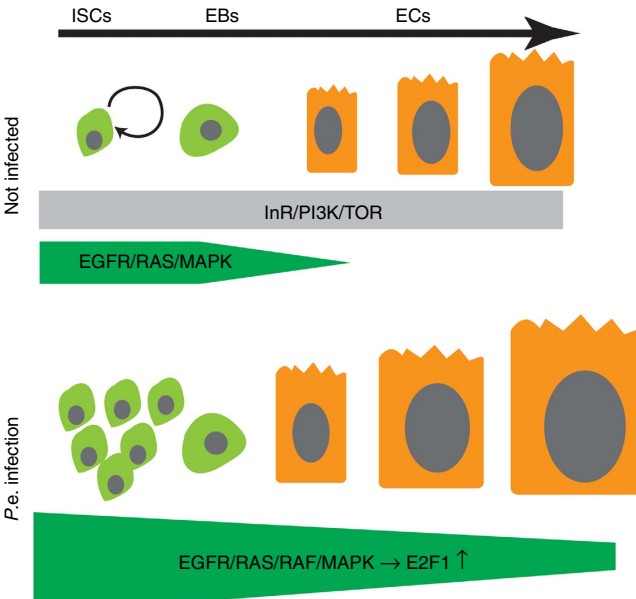

**Figure 7 | Model illustrating the relative influence of EGFR/Ras/Raf/ MAPK and InR/Pi3K/TOR signalling on ISC proliferation and EB/EC endoreplication.** Upper panel: Under normal physiological conditions, both EGFR/MAPK and InR/Pi3K/TOR pathways act in progenitor cells to control their proliferation and growth. But only the InR/Pi3K/TOR pathway, and not the EGFR/MAPK pathway, is active in ECs to promote their growth with endoreplication. Lower panel: During infection-induced regeneration, EGFR/MAPK signaling is strongly induced in progenitor cells and inherited by their progeny. This results in hyperproliferation of ISCs and hypertrophy and higher ploidy in ECs. EGFR/MAPK signaling post-transcriptionally up-regulates E2f1 to induce this EC endorelplication.

UAS-Notch RNAi, UAS-Ras RNAi and USA-Raf RNAi (NIG, Japan and VDRC, Austria). The following mutants were used: FRT82B EGFR$^{CO}$, FRT82B Ras$^{\Delta 40b}$, FRT82B InR$^{339}$, FRT 82B Dp110$^A$ and FRT40A TOR$^{\Delta P}$. MARCM stocks are as follows: FRT42D: yw hsflp UAS-GFP tubGal4; FRT42D tubGal80; +, FRT40A: UAS-CD8-GFP hsflp; FRT40A tubGal80; tubGal4. FRT82B: yw hsflp UAS-GFP tubGal4; +, FRT82B tubGal80. FRT80B: yw hsflp UAS-GFP tubGal4; +, FRT80B tubGal80. Midgut Gal4 drivers are as follows: MyoIAGal4$^{ts}$: w, Myo1AGal4 tubGal80$^{ts}$ UAS-GFP; +, esgGal4$^{ts}$: w, esgGal4 tubGal80$^{ts}$ UAS-GFP. Su(H)GBE$^{ts}$: w, Su(H)GBEGal4 UbiGal80$^{ts}$ UAS-GFP. esg F/O: yw, esgGal4 tubGal80ts UAS-flp UAS-GFP; and Act>CD2>Gal4.

*Drosophila* stocks were maintained by standard methods at 25 °C. For experiments using *Gal80$^{ts}$*, flies were raised at 18 °C and then shifted to the restrictive temperature (29 °C) at 3–5 days after eclosion. For generation of mosaic mutant clones, 3—5-day-old flies were heat shocked once for 30 min at 37 °C. For generation of UAS-transgene overexpression clones, adult flies were shifted at 29 °C for various times. To control the timing of UAS-transgene overexpression with the *esg* Flip/Out system, we shifted 3–5-day-old female flies to 29 °C to generate flip-out clones, and 1 day after clone induction we fed them *P.e.* bacteria dissolved in a 5% sucrose solution. To overexpress $Ras^{V12S35}$ in $InR^{-/-}$, $Pi3K^{-/-}$ or $TOR^{-/-}$ cells, we used the MARCM system[60].

**Immunofluorescence and image analysis.** Midguts were dissected in phosphate-buffered saline (PBS) and fixed for 30 min at 25 °C in 4% paraformaldehyde in PBS. Isolated intestines were blocked for 30 min at 25 °C in PBS/0.1% Triton X-100/10% NGS. Primary antibodies were used in the following dilutions: mouse anti-EGFR (1:200; ABgent), chicken anti-GFP (1:500; Invitrogen), rabbit anti-PH3 (1:2,000; Millipore), mouse anti-pERK42/44 (1:500; Cell Signaling) and mouse anti-PDM1 (1:1,000; gift from Xiaohang Yang, National Biotechnology Institute, Singapore). Secondary antibodies conjugated with Alexa488 568 or 633, purchased from Invitrogen, were used at a dilution of 1:1,000. DNA was visualized with 4,6-dia-midino-2-phenylindole (DAPI; 0.5 mg ml$^{-1}$; Sigma) diluted 1:2,000. For EdU labeling, 2 mg ml$^{-1}$ EdU (Sigma) in PBS was added to the yeast–molasses-based food. After 8 h at 29 °C, flies were dissected and processed as above. Samples were mounted in Vectashield (Vector Laboratories) and imaged with a Leica SP5 confocal microscope using 40× oil and 63× oil objectives (imaging medium: immersion oil Type F obtained from Leica). The imaging was performed at room temperature. Images were processed in Image J, Photoshop CS5 (Abode) and Illustrator CS4 (Adobe) for image merging and resizing.

**Flow cytometry.** Midguts of 15 females per experiment were dissected in PBS supplemented with 1% fetal calf serum. Midguts were washed three times in calcium-free PBS and incubated in 500 ml of 5% collagenase and 0.1% trypsin with intermittent vortexing at room temperature for 50 min. The cell suspension was centrifuged at 3,000 r.p.m. for 10 min. Cell pellets were then resuspended with 500 ml nuclear purification buffer and vortexed 3–5 times. Suspensions were centrifuged once more and then nuclei were resuspended in 500 ml PBS with 0.5 µg ml$^{-1}$ DAPI. The nuclear suspension was passed through a 40 mm nylon filter into a BD FACS tube, incubated at room temperature for 20 min, washed once in calcium-free PBS and stored on ice. A flow cytometer (FACS CantoII; Becton Dickinson) determined midgut cell ploidy by FACS analysis of DAPI-stained nuclei preparations with excitation at 405 nm for DAPI stain and at 488 nm for GFP.

**Quantification of DNA.** The DNA content of ECs for each genotype was mea-sured. DAPI-stained nuclei were imaged in *Z*-stacks using a Leica SP5 confocal microscope, using low laser paper to avoid saturation of the detectors. Every nucleus was scanned completely from the top to button focal plane, using *Z*-stack steps of 0.25 µm. We then calculated the average fluorescent intensity of each layer using Imaging J. Finally, signal from all the planes was merged together. The total florescent intensity for each nucleus was determined as the sum of the integrated average fluorescent intensity of each layer.

**Western blots.** A total of 15 midguts were dissected from each control, $Ras^{V12S35}$ or $Raf^{GOF}$ overexpression groups. Tissuelyser was applied to lysate the midguts. Total proteins from these guts were then extracted within RIPA buffer. Rabbit anti-E2f1 antibody (1:1,000 dilution) was used for this blot. Anti-GFP antibody was purchased from Invitrogen (1:5,000 dilution). The uncropped western blot can be found in Supplementary Fig. 8.

**Statistical analyses.** Two-tailed unpaired *t*-tests, assuming equal variances, were performed for all statistical analyses. $P < 0.01$ was considered statistically significant for all analyses.

**Data availability.** The authors declare that all data supporting the findings of this study are available within the article and its Supplementary Information files or from the corresponding author on reasonable request.

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

## Acknowledgements

We acknowledge Norman Zielke, Jerome Korzelius and members of the Edgar lab for help and discussion. Special thanks to Monika Langlotz of the ZMBH Flow Cytometry Facility. Thanks to Gen Lin at EMBL for helping with RNA-sequencing data analysis. We thank the ZMBH Imaging Facility. This work was supported by ERC AdG 268515, DFG SFB 873 and DKFZ funding to B.A.E.

## Author contributions

J.X. and B.A.E conceived of the study and designed the analysis. J.X., P.Z., Y.J. and H.R. performed the experiments. J.X., J.B. and B.A.E. wrote the manuscript.

## Additional information

**Competing interests:** The authors declare no competing financial interests.

