## [Peer Review File · Nature Communications]

Reviewer #1 (Remarks to the Author)

Xiang et al. investigate the role of cellular enlargement via endoreplication in the regenerative response to P.e. infection. The authors had previously observed increased enterocyte cell (EC) size in the midgut when the intestine was damaged by killing ECs with expression of reaper or when regeneration was blocked upon ISC killing by p53 expression (Jiang, Cell, 2009). Here they investigate the role of this increased EC size in promoting viability after damage and the cellular pathways that drive the increased nuclear and cytoplasmic size. They provide evidence that EGFR/Ras/Raf/MAPK specifically promote endoreplication in stress contexts and act independently of Insulin receptor and TOR signaling.

This is an interesting study, that, in general, is well performed. It provides novel insight into an interesting phenomenon of compensatory cell growth that will be of broader interest. However, I would suggest that a few points be strengthened prior to publication:

1. A major claim of the paper is that the cell enlargement in response to stress is beneficial for the fly. The main data in support of this notion are presented in Fig. 1G and use the Su(H)GBE-Gal4 to express Geminin or Pol alpha IR which should block endoreplication. The authors interpret a decrease in viability to indicate that "that compensatory cell growth driven by polyploidization is an essential aspect of the stress response." Under P.e. infection conditions, a large number of new EC cells will be needed, therefore blocking EC endoreplication in general would be expected to result in decreased viability. Therefore, from this experiment, it is not clear whether it is the compensatory endoreplication per se or whether it is endoreplication in general which would be presumably required for proper EC cells. The authors should try to test directly the function of the enhanced endoreplication leading to higher ploidy on the regenerative response. This could be done by specifically knocking down EGFR with Su(H)GBE-GAL4 or a Su(H)GBE-GAL4 flip-out strategy. They should assess the effect on ploidy in response to P.e. and the effect on fly viability.

2. I was a bit confused by the data in Fig. S3: When Ras or Raf is expressed in ECs, it was found to be sufficient to promote Edu incorporation in 90% of ECs. However, the effect "was small" on increased ploidy. The authors speculate that they are past a point at which Ras can be effective, but if so, then why were 90% of ECs positive? Minimally, this needs a better explanation in text as to what could be happening.

3. The authors mention the role of Notch in promoting endoreplication in other tissues in their introduction, but do not really come back to the function of Notch in their system. Could they test whether Notch is indeed required here? Possibly by knocking down Notch after it has first been activated in EBs using Su(GBE)Gal4, they could assess whether ongoing Notch signaling would be important for compensatory hypertrophy.

4. In many instances, which cells are being quantified for DNA/nuclear size is unclear. This is especially problematic in the mutant contexts. It would make sense to only look at EC cells (excluding DI+, Pros+ etc). For example, in Figure 3E- in the Ras mutants, how are ECs being defined? Are they scoring all cells or just ECs? Similarly, in Fig-5F- in the E2F mutants- how are ECs being defined? How can they exclude that the effect they see is not simply blocked ISC division and they are therefore scoring ISCs? This is important as they are making a claim on endoreplication in ECs. A role on endoreplication could be distinguished using SuH>E2F IR. In addition, it would be helpful to see what a normal diploid score is for DNA/nuc in at least one figure for comparison.

A related point: in Fig. 3E, the data show a significant decrease in Ras mutant conditions compared to controls, yet they say that this is "slight" and then go on to conclude that EGFR/Ras has no effect. To me this logic is a bit strange. Is this in fact due to the cell populations they are scoring- mixing ISCs and ECs in wild-type and only single ISCs in the Ras mutants?

5. In Fig. 4F, it would be helpful to have the RasV12 and Raf GOF data alone in same experiment. It is unclear whether RasV12 and Raf GOF can drive even higher endoreplication in this context and therefore the effect in combination with PI3K would be the additive effects of increasing and reducing. These controls should be included.

6. For comparison, it would be helpful to see a control in Fig 2 for EGFR expression in uninfected guts.

Minor points:

1. Some sentences in the intro are a bit awkward and there are some typos:

"Cellular adaptation is a cell change in response to its' living environmental changes."

"The endodermal portion of the Drosophila intestine, called the midgut, as a good cell adaption model, is maintained by intestinal stem cells (ISCs)."

etc

In Fig 2K- nucleus, not "neclous"

2. Methods say that Hoechst was used whereas DAPI is indicated in text, quantification and figures. Please fix.

Reviewer #2 (Remarks to the Author)

This paper describes cellular compensatory hypertrophy following gut epithelial damage. During this process, endoreplication is nearly exclusively in new enterocytes and results from EGFR signaling. Though Insulin-receptor signaling required for homeostatic maintenance of this post-mitotic tissue, it is not needed for ectopic endoreplication in response to infection stress. Finally, the manuscript shows that Ras/Raf signaling upregulates E2F1 post-transcriptionally, so as to induce endoreplication. Overall, the experiments are all very well designed, and are aimed at answering specific questions. The data look reliable and the phenotypes are quite obvious to see in the images provided.

Despite all the merits, there are several issues need to be addressed.

1. The role of E2f1 remains preliminary and somewhat correlative. They show that E2F1 is translationally (or post-translationally) activated by Ras/MAPK, but do not show how. They discuss that it may be through Myc, but whether the activation of E2f1 is via Myc has not been tested, and probably important to show in this study to make the story complete.

2, The arrangement and distribution of figures in the paper need to be organized in a more sequential order, and flow with the text, rather than switching between figures in the paper and in the supplemental constantly. It is very difficult to follow if the figures are arranged the way they are now.

Also, if the images in each figure can be arranged in the order that they are discussed in, it will make a lot more sense.

3. The conclusion that EGFR/MAP is transcriptionally down-regulated during EB-EC differentiation and EC maturation is not supported.

4. In Fig. 2E-E': Either call the genotype λ TOP OR EGFR Δ CT, not switch between one and the other.

Page 6 It says Jak/Stat in one of the later sentences. Please correct to say JAK/STAT.

5. Page 9 The paragraph deals with cell volumes, but the last line says cell mass. Is the mass in question, or the volume?
6. Fig 1 G + Page 10 paragraph 2: If there can be an image showing that there was no compensatory hypertrophy with either Geminin, or DPα overexpression, it would be more easily understandable than showing the percent survival. With the data provided now linking survival to endoreplication, it is not direct and appears correlative. %Survival is reported only in this study and it seems rather out of place.
7. Fig 2 Control for αEGFR levels can be provided as well, since one for α-pERK has also been provided.
8. For statistic analyses, the number of samples counted should be noted in the manuscript. Also, statistical analysis is needed to compare InR-/- clones and wildtype clones. Moreover, the apoptotic effect should be assessed in InR-/- clones.
9. how about other components derived from intestinal stem cells. Would they be affected by different genetic manipulations?
10. Why is InR signaling required for non-stress homeostasis, but not for induced endoreplication? What would happen if knocking down InR while overexpressing EGFR?
11. How is endoreplication achieved by EGFR signaling? Can they test some targets of EGFR that are related to endoreplication?
12. More EdU incorporation only means Ras can induce endoreplication events, not necessarily indicates Ras accelerates it. They can consider to measure the doubling time. Further, they need to use other components and RNAi's in EGFR signaling to confirm the involvement of this pathway.
Evidence to support that Geminin suppresses endoreplication should be included.
13. There are several invalid statement throughout the context, without any reference, e.g. "growth in most of the larva?s tissues is driven primarily by increases in cell size rather than cell number". Several important references are missing.
In the context of "compensatory cell growth" in Drosophila adult intestines, the paper by Kollahgar et al. (Dev Cell. 2015 34:297-309) showed an important example. Kollahgar's paper has shown that the proliferation rate of wild-type stem cells is accelerated in Minute vs. wild-type cell competition through chronic JNK-activation-induced JAK/STAT signaling upregulation. The paper should be cited appropriately
14. The involvements of JNK and JAK/STAT signaling pathway in the compensatory hypertrophy of ECs should be examined.
15. there are several noticeable grammatical errors and awkward wording throughout the context.
Eg. Page 5: "into two the major classes" should be "two of the major classes"
P10: "this data" should be "these data"

Reviewer 1

1. A major claim of the paper is that the cell enlargement in response to stress is beneficial for the fly. The main data in support of this notion are presented in Fig. 1G and use the Su(H)GBE-Gal4 to express Geminin or Pol alpha IR which should block endoreplication. The authors interpret a decrease in viability to indicate that " that compensatory cell growth driven by polyploidization is an essential aspect of the stress response." Under P.e. infection conditions, a large number of new EC cells will be needed, therefore blocking EC endoreplication in general would be expected to result in decreased viability. Therefore, from this experiment, it is not clear whether it is the compensatory endoreplication per se or whether it is endoreplication in general which would be presumably required for proper EC cells. The authors should try to test directly the function of the enhanced endoreplication leading to higher ploidy on the regenerative response. This could be done by specifically knocking down EGFR with Su(H)GBE-GAL4 or a Su(H)GBE-GAL4 flip-out strategy. They should assess the effect on ploidy in response to P.e. and the effect on fly viability.

Answer:

This is a logical criticism, and in fact while our results confirm the importance of EGFR for endoreplication in general, we can't actually provide clear evidence that the additional cell enlargement (above normal cell size) or hyperpolyploidy that occurs during rapid regeneration after infection contributes to this fitness benefit. Experimentally, it is very difficult to separate additional endoreplication from normal endoreplication, since they occur in the same developmental time window (EB→EC transition) and are both driven by the same mechanisms (EGFR signaling).

To try to address the distinction, we've done the experiment the reviewer suggests, namely depleting the EGFR in EBs using the EB-specific Su(H)GBE-Gal4 driver. Unfortunately, no significant reduction of endoreplication was observed. We believe this is because the Su(H)GBE promoter is activated only for a short time in developing EBs, and so the EGFR-RNAi may not be effective in this brief interval. In fact our RNAseq and protein expression data indicate that *EGFR* is produced primarily in activated ISCs, and that *EGFR* protein is inherited by EBs and ECs. Therefore, expressing EGFR-RNAi in EBs may be too late to affect levels of this receptor during endoreplication.

In response to the reviewer's comment, we have revised the Abstract, Introduction, Results and Discussion of our paper so as not to imply that extra-endoreplication and compensatory hypertrophy, as opposed to endoreplication and cell growth in general, is required for fitness. We know that endoreplication is required (Fig 1H), but the importance of the compensatory endoreplication and cellular hypertrophy observed during the regenerative response is technically very difficult to address, and so the idea that it is important remains conjecture. The revised text is now clear on this issue.

2. I was a bit confused by the data in Fig. S3: When Ras or Raf is expressed in ECs, it was found to be sufficient to promote Edu incorporation in 90% of ECs. However, the effect "was small" on increased ploidy. The authors speculate that they are past a point at which Ras can be effective, but if so, then why were 90% of ECs positive? Minimally, this needs a better explanation in text as to what could be happening.

Answer:

Forcing *Ras* or *Raf* expression in EBs and pre-ECs (Su(H)-Gal4 driver) induced much higher ploidy cells. This confirmed that *Ras* or *Raf* accelerates endoreplication. However, the EC-specific *Myo1A-Gal4* driver is activated rather late in EC maturation. Our results indicate that these fully differentiated mature ECs have less capability for overgrowth than EB or pre-ECs. This conclusion is corroborated by the results in Fig 1E,F, showing that mature ECs cannot re-enter the endo-cell cycle in response to *P.e.* infection. Nevertheless, as we show in Fig S3, the re-replication of many *Myo1A*⁺ cells still can be accelerated by *Ras* or *Raf* activity. We expect that the ECs that respond are probably the youngest ECs, but that they still have a very limited ability to replicate, or lose their ability to replicate as they mature during the course of the experiment. This limited responsiveness would explain why they can be EdU incorporation positive, but still don't have detectably enlarged nuclei. They probably do one or fewer extra rounds of replication. It is also the case that in this experiment (w/ *Myo1A*, Fig S2) *Ras* or *Raf* was expressed for only one day, and not three, as with the Su(H)-Gal4 driver, so there was less time for ploidy increases. We don't know exactly why mature ECs would become refractory to *Ras/Raf* signaling, but we offer a few ideas in the Discussion (page 23). For instance, it may be because mature ECs have lower levels of MEK and ERK (see new Fig S2).

3. The authors mention the role of Notch in promoting endoreplication in other tissues in their introduction, but do not really come back to the function of Notch in their system. Could they test whether Notch is indeed required here? Possibly by knocking down Notch after it has first been activated in EBs using Su(GBE)Gal4, they could assess whether ongoing Notch signaling would be important for compensatory hypertrophy.

Answer:

This is a very logical suggestion, as Notch signaling is required for EC differentiation in midgut regeneration, and therefore for endoreplication. However technically, it will be very difficult to separate the effects of Notch on EC differentiation and endoreplication. Because Notch-manipulation experiments from multiple labs show a 1:1 correspondence between EC differentiation and polyploidy, we don't think the experiment the reviewer suggests would be informative. An in-depth analysis of Notch target genes in EBs could be informative, but we think endeavoring to do that would be well beyond the scope of this focused study. Of note, we have done *P.e.* oral infection to the flies after

Notch depletion with RNAi driven by *esg-Gal4*. This unpublished data shows that blocking Notch activity, and also the switch to endoreplication, strongly reduces the flies' survival in stress conditions. This is consistent with the idea that Notch is required for the switch to endoreplication and the production of large ECs.

*4. In many instances, which cells are being quantified for DNA/nuclear size is unclear. This is especially problematic in the mutant contexts. It would make sense to only look at EC cells (excluding *DI+*, *Pros+* etc). For example, in Figure 3E- in the Ras mutants, how are ECs being defined? Are they scoring all cells or just ECs? Similarly, in Fig-5F- in the E2F mutants- how are ECs being defined? How can they exclude that the effect they see is not simply blocked ISC division and they are therefore scoring ISCs? This is important as they are making a claim on endoreplication in ECs. A role on endoreplication could be distinguished using *SuH>E2F IR*. In addition, it would be helpful to see what a normal diploid score is for DNA/nuc in at least one figure for comparison.*

A related point: in Fig. 3E, the data show a significant decrease in Ras mutant conditions compared to controls, yet they say that this is "slight" and then go on to conclude that EGFR/Ras has no effect. To me this logic is a bit strange. Is this in fact due to the cell populations they are scoring- mixing ISCs and ECs in wild-type and only single ISCs in the Ras mutants?

Answer:

This is an important question. We also considered this issue when we were scoring the mutant clones. In these experiments we scored all cells, not just ECs as the reviewer suggests. While scoring only ECs would have been ideal, it was not very practical because the anti-PDM1 antibody that marks ECs is in limited supply and doesn't always give clean results. Nevertheless, we did use PDM-1 staining in some experiments to confirm, as previously demonstrated by Jiang et al. (Cell Stem Cell 2011), that altering EGFR/RAS activity doesn't affect EC differentiation. See the new Supl. Fig 4 J-J".

Regarding the question to Fig.5, E2F mutant cells lost their ability to divide or enter endoreplication. This is expected as E2F1 controls many cell cycle genes in most eukaryotic cells, and also is essential to maintain endoreplication. We did stain these guts with anti-PDM1 antibody. Some of the E2F-mutant, diploid cells are PDM1 positive, indicating that they have undergone EC differentiation, and that E2F1, like Ras, is not required for differentiation (see new Suppl. Fig. 7 A-A").

Regarding the suggested experiment with *Su(H)GBE-Gal4>UAS-E2F1^{RNAi}*, we have performed this experiment, but found that the supposed suppression of E2F activity had little detectable effect. We ascribe this to technical shortcomings with the experiment, namely that using this driver to express RNAi gives too little expression, too late, to be effective in blocking endoreplication. This is the same

issue noted above in response to question #1. Therefore we don't include this negative data here. Nevertheless, our observation that E2F1 mutant cells in clones can be Pdm1+ but remain diploid confirms that E2F1 is essential for EC endoreplication.

5. In Fig. 4F, it would be helpful to have the RasV12 and Raf GOF data alone in same experiment. It is unclear whether RasV12 and Raf GOF can drive even higher endoreplication in this context and therefore the effect in combination with PI3K would be the additive effects of increasing and reducing. These controls should be included.

Answer:

The revised Fig.4 F now includes the *Raf*^{GOF} alone clone data. Fig. S5 has the *Ras*^{V12S35} alone clone data. These data all come from the same experiment, so the data are directly comparable. Although the Raf and Ras inductions alone gave slightly more growth than in the presence of Tor, Pi3K or InR mutations, the data do not suggest simply additive effects. Rather, Raf and Ras seem to bypass the loss of Pi3k/Tor activity. To avoid showing redundant data in the main figures, we think this is the optimal layout of the data. In the revised text (Fig legend 4, p37) we note that the data in Fig S5 were obtained in parallel and are directly comparable.

6. For comparison, it would be helpful to see a control in Fig 2 for EGFR expression in uninfected guts.

Answer:

We have added the control picture to Fig 2. (Page 46)

Reviewer 2

1. The role of E2f1 remains preliminary and somewhat correlative. They show that E2F1 is translationally (or post-translationally) activated by Ras/MAPK, but do not show how. They discuss that it may be through Myc, but whether the activation of E2f1 is via Myc has not been tested, and probably important to show in this study to make the story complete.

Answer:

This is an insightful question. Our work showed that Ras/Raf signaling promotes EC endoreplication, that E2F1 is required for and capable of promoting endoreplication, and that Ras/Raf enhances E2F levels and activity. Although we don't determine the mechanism of post-transcriptional upregulation of E2F, we do

feel we've provided much more than a correlative argument that Ras regulation of E2F drives endocycling. This set of results suggests causality.

We did wonder whether *Myc*, as an important factor in cell growth, may be involved in this control. Therefore in response to the reviewer's query we overexpressed cMyc in EBs using Su(H)GBE-Gal4. However *Myc* overexpression did not drive endoreplication in this context. We did not observe larger cells with higher ploidy than wild-type control cells. In addition we generated *myc* null clones in midguts. All the cells in the *Myc* null mutant clones were diploid and small in size, and the clones were very small (1-2 cells). These results indicate that *Myc* is required for ISC proliferation and EB endoreplication, but insufficient to induce endoreplication and cell growth in EBs (see new Supl. Fig7 B-C"). In addition to this new data on *Myc*, we also now include reference to a paper and our unpublished data that indicate that S6K is not an important growth effector in the fly midgut (see Discussion, p25).

Although we'd like to add further data on the mechanism of growth and E2F control by Ras/Raf, we don't have anything more to report, and the experiments that are in progress on this topic will take many months to complete. We hope the reviewer will agree that demonstrating the sufficiency and requirement of E2F for endoreplication, and its post transcriptional control by Ras/Raf, are significant findings that add to the value of this paper.

2. The arrangement and distribution of figures in the paper need to be organized in a more sequential order, and flow with the text, rather than switching between figures in the paper and in the supplemental constantly. It is very difficult to follow if the figures are arranged the way they are now. Also, if the images in each figure can be arranged in the order that they are discussed in, it will make a lot more sense.

Answer:

It is a good suggestion. We have tried our best to organize the figures. Some out-of order data has been moved to align better with the text (e.g. Fig 2). Because of the limitation of space, we've had to move some pictures from the main figures to the supplement. For example, as *Ras*^{-/-} and *PI3K*^{-/-} represent the MAPK and InR/PI3K signaling respectively, *EGFR*^{-/-} and *InR*^{-/-} clone data have been moved to supl. 4.

3. The conclusion that EGFR/MAP is transcriptionally down-regulated during EB-EC differentiation and EC maturation is not supported.

Answer:

The conclusion that EGFR/MAP Kinase is transcriptionally down-regulated during EB-EC differentiation is now supported by RNA-Seq data in Fig. S2, an new figure.

4. In Fig. 2E-E': Either call the genotype λ TOP OR EGFR^{ACT}, not switch between one and the other. Page 6 It says Jak/Stat in one of the later sentences. Please correct to say JAK/STAT.

Answer:

We have replaced λ TOP with EGFR^{ACT} in the legend for Fig 2 (page 34). On page 6, Jak/Stat has been corrected.

5. Page 9 The paragraph deals with cell volumes, but the last line says cell mass. Is the mass in question, or the volume?

Answer:

Our measurements by FACS give values proportional to cell volumes (as stated), but we assume that cell mass is proportional to volume in these experiments, and it is mass that actually interests us. Since the last line (paragraph 1, p8) is a conjecture, we think it's OK to us "mass" here, as the important parameter.

6. Fig 1 G + Page 10 paragraph 2: If there can be an image showing that there was no compensatory hypertrophy with either Geminin, or DP α overexpression, it would be more easily understandable than showing the percent survival. With the data provided now linking survival to endoreplication, it is not direct and appears correlative. %Survival is reported only in this study and it seems rather out of place.

Answer:

We understand the reviewer's concern. The relevant data is shown in Fig 2, I-K. Although we didn't use *P.e.* infection, we performed a similar experiment in which we overexpressed both Ras^{V12S35} and *Geminin* in EBs using Su(H)GBE-Gal4. In this experiment, we observed the Ras-induced endoreplication was strongly inhibited by Geminin. In the revision, we now cross reference this experiment where we discuss the survival data from Fig 1.

7. Fig 2 Control for α EGFR levels can be provided as well, since one for α -pERK has also been provided.

Answer:

We have added this control picture to Fig 2.

8. For statistic analyses, the number of samples counted should be noted in the manuscript. Also, statistical analysis is needed to compare InR^{-/-} clones and wildtype clones. Moreover, the apoptotic effect should be assessed in InR^{-/-} clones.

Answer:

We have added the number of samples in the figure legends as suggested by the reviewer. (see pages 34-43). Statistical analysis for $InR^{-/-}$ is shown in Fig. 3j (see page 47).

Regarding the $InR^{-/-}$ clones, in every gut, the number of $InR^{-/-}$ clones was much smaller than wild-type clones. It is highly possible that deletion of InR resulted in cell loss. Although we could count it and get this information, a defect in stem cell maintenance is expected, and this point is not related with the main conclusion we would like to make in this manuscript.

9. how about other components derived from intestinal stem cells. Would they be affected by different genetic manipulations?

Answer:

This is a common problem for all of the genetic manipulation in *Drosophila*. We would believe that other components have been affected. But our work in this manuscript focuses on the components of MAP Kinase signaling and how they affect compensatory hypertrophy.

10. Why is InR signaling required for non-stress homeostasis, but not for induced endoreplication? What would happen if knocking down InR while overexpressing $EGFR$?

Answer:

This is a very interesting question. We do not know why InR becomes dispensible under stress. Based on our cell type specific RNA-Seq data, *P.e.* stress did not induce insulin-like peptides or InR in midguts. However, under this stress, $EGFR$ ligands were strongly induced. Meanwhile, after infection of guts carrying $EGFR$ null clones, cells lost their capability to undergo endoreplication induced by stress. As we explain in the Discussion, $EGFR$ signaling appears to supplant InR signaling during stress-induced regeneration. How $EGFR/RAS/RAF/MAPK$ signaling bypasses the requirement for $Pi3K$ and TOR is a very interesting question for future studies.

11. How is endoreplication achieved by $EGFR$ signaling? Can they test some targets of $EGFR$ that are related to endoreplication?

Answer:

We have tested some of the most likely $EGFR$ targets, including *Cic*, *Pointed*, *Ets21c*, *Myc*, *Jun*, *Fos* and *S6K* for the ability to promote endoreplication in midguts, like *Ras* or *Raf*. These genes were overexpressed in EBs and pre-ECs using *Su(H)GBE* for 3 or 5 days. Unfortunately, we did not observe that any one of them could promote endoreplication. Furthermore, we have generated *Myc* null clone cells in midguts. These cells could not undergo endoreplication. All of them arrested with diploid status. These data indicate that these factors cannot

drive endoreplication alone, though Myc at least is required for it. The most we can conclude at this point is the effect of *EGFR* is MAPK-dependent, and this is clearly noted in the paper.

12. More EdU incorporation only means Ras can induce endoreplication events, not necessarily indicates Ras accelerates it. They can consider to measure the doubling time. Further, they need to use other components and RNAi's in EGFR signaling to confirm the involvement of this pathway. Evidence to support that Geminin suppresses endoreplication should be included.

Answer:

It's correct that EdU incorporation cannot be used to infer an acceleration of S-phases. However in many of our experiments we actually measure DNA amounts (ploidy) relative to controls after a fixed experimental duration. If more DNA is generated in experimental samples than in controls in the same amount of time, then the endocycle has been accelerated. This kind of result is seen in Figs 1,2,3,4, and 5.

The evidence that Geminin suppresses endoreplication is shown in Fig 2I-L.

13. There are several invalid statement throughout the context, without any reference, e.g. "growth in most of the larva's tissues is driven primarily by increases in cell size rather than cell number". Several important references are missing.

In the context of "compensatory cell growth" in Drosophila adult intestines, the paper by Kolahgar et al. (Dev Cell. 2015 34:297-309) showed an important example. Kolahgar's paper has shown that the proliferation rate of wild-type stem cells is accelerated in Minute vs. wild-type cell competition through chronic JNK-activation-induced JAK/STAT signaling upregulation. The paper should be cited appropriately

Answer:

That is a good suggestion. We have cited the Kolahgar paper as ref [30], on Page 7.

14. The involvements of JNK and JAK/STAT signaling pathway in the compensatory hypertrophy of ECs should be examined.

Answer:

We have overexpressed *Hop^{Tum-I}* (an activated allele of JAK) in ECs for 7 days. This ectopic expression of activated JAK did promote endoreplication, though not nearly as strongly as Ras or Raf. However, since the effect was relatively weak, we did not generate the loss of function of JAK/STAT in midgut cells and then analyze DNA endoreplication. Therefore, we cannot conclude or exclude that JAK/STAT is involved in compensatory hypertrophy. As this paper is focused on EGFR signaling, we've omitted this rather inconclusive data.

Regarding JNK signaling, we have overexpressed cJun or cFos in EBs. But neither cJun nor cFos could not promote endoreplication. Since the effects of EGFR/Ras signaling on endoreplication were strongest, we focused our efforts in this paper on learning how this system acts.

15. there are several noticeable grammatical errors and awkward wording throughout the context.

Eg. Page 5: "into two the major classes" should be "two of the major classes"

P10: "this data" should be "these data"

Answer:

We have corrected these errors (page 5) and (page 10), as well as many others. Thanks for pointing them out.

Reviewer #1 (Remarks to the Author)

The revised manuscript addresses a large number of my previous criticisms and I now find suitable for publication.

However, I would just like to mention that it seems to me in comparing my older version and the newer version, that several times the author's claimed that new figures were added and, from what I can see, this seems not to be the case. For example, in response to my question 5, they say that they now include Raf and Ras clone data in Fig 4 and S5. They did not change these aspects of the figures; for me it is OK as they come from the same experiment and are comparable; so my point is more or less addressed. Similarly, in response to Reviewer 2's point 3, they state that Fig S2 is a new figure, though I do not believe that it has changed. That being said, the previous figure does support the claim that EGFR components transcripts are changing during differentiation. They could have simply referred to the figures...

Reviewer #2 (Remarks to the Author)

My concerns have been adequately addressed.

Suggest the authors thoroughly check the manuscript for typos and other errors.

For example:

Typo in page 7 , "acceleratinXiang"

Repeated words in page 19 "we found that that myc mutant cells"